ecology/behaviour/environmental science

real-world heart rates, data logging, everyday sounds, linear mixed-effects models, hearing loss, hearing aids

**Author for correspondence:**
Jeppe H. Christensen
e-mail: jych@eriksholm.com

# The everyday acoustic environment and its association with human heart rate: evidence from real-world data logging with hearing aids and wearables

Jeppe H. Christensen[1,2], Gabrielle H. Saunders[3], Michael Porsbo[2] and Niels H. Pontoppidan[1,2]

[1]Eriksholm Research Centre, Snekkersten, Denmark
[2]Oticon A/S, Smørum, Denmark
[3]Manchester Centre for Audiology and Deafness, School of Health Sciences, University of Manchester, Manchester, UK

JHC, 0000-0002-6666-8869

We investigate the short-term association between multidimensional acoustic characteristics of everyday ambient sound and continuous mean heart rate. We used in-market data from hearing aid users who logged ambient acoustics via smartphone-connected hearing aids and continuous mean heart rate in 5 min intervals from their own wearables. We find that acoustic characteristics explain approximately 4% of the fluctuation in mean heart rate throughout the day. Specifically, increases in ambient sound pressure intensity are significantly related to increases in mean heart rate, corroborating prior laboratory and short-term real-world data. In addition, increases in ambient sound quality—that is, more favourable signal to noise ratios—are associated with decreases in mean heart rate. Our findings document a previously unrecognized mixed influence of everyday sounds on cardiovascular stress, and that the relationship is more complex than is seen from an examination of sound intensity alone. Thus, our findings highlight the relevance of ambient environmental sound in models of human ecophysiology.

## 1. Introduction

Humans constantly face a multifaceted soundscape. Some content is by choice (e.g. listening to music), while other is not (e.g. traffic

noise). Despite environmental sounds being an integral part of everyday human life, there is very little knowledge about how acoustic aspects of the soundscape encountered in everyday life, beyond that of intensity, affect the human body [1]. There is, however, increasing recognition that sounds classified as 'noise' have harmful effects on the human auditory [2,3] and cardiovascular system [4,5] despite being at intensity levels below those known to cause physical damage. A recent report from the WHO summarized the biological mechanism of cardiovascular reactions to noise [6]. In brief, it suggested that auditory noise causes both a direct (i.e. through subcortical connections of the brain) and/or indirect (involving projections from the auditory thalamus to the auditory cortex) stress reaction by disrupting the autonomic nervous system (ANS) balance due to elevated activity in the sympathetic nervous system branch and reduced activity in the parasympathetic nervous system branch [6–10], both of which control everyday fluctuations in heart rate (HR). Elevation in sympathetic control leads to slow (approx. 5 s delay) increases in HR [11], while elevated parasympathetic control leads to fast (within milliseconds) decreases in HR [12,13]. In the case of noise exposure, the elevation of the sympathetic control can have acute effects that lead to momentarily elevated HR and blood pressure [14] and can cause long-term damage, such as elevating the risk of heart disease due to sustained periods of hypertension [15]. What is more, effects of noise are not always perceivable, as illustrated in a study [16] in which clerical workers were exposed to simulated open-office noise. Compared with a control group they showed increased levels of adrenaline, indicative of elevated sympathetic levels, but did not report higher stress than the control group. Thus, to fully understand the impact of the everyday acoustic environment on the human body, subjective reporting of stress needs to be supported by objective measures such as cardiovascular reactions toward changes in the environment.

In a recent laboratory study, Shoushtarian *et al.* [17] documented a causal link between short-term noise exposure and acute cardiovascular system reactions. They found that relative to complete silence, noise segments presented at 40 and 15 dBA led to a decrease in HR, while those presented at levels of 65 and 90 dBA led to significant increases in HR. These changes were seen within 2–5 s post exposure. This timing is in line with the dynamic response of sympathetic ANS modulation of HR [12], rather than the faster acting parasympathetic control [11], suggesting that, indeed, changes in the intensity of acoustic noise induce a 'fight and flight' stress response in the sympathetic nervous system branch [5]. Pairwise comparisons of the changes revealed significant differences in HRs between all comparison levels except for between 65 and 90 dBA, suggesting a possible ceiling effect. The authors interpreted the 2% reduction in mean HR from baseline for the 15 dBA noise to be due to a fast-acting orienting response elicited by the onset of low intensity noise [18], and their finding suggests that noise can have diverse effects on the ANS depending on its intensity.

Diverse ANS reactions to sound have also been linked to the type of sound, psychological evaluation and task-related listening. For example, Sim *et al.* [19] reported that type of noise (e.g. traffic noise, background noise, speech noise) when presented at equal intensity differentially impacts heart rate variability (HRV). They argued that, besides activating a sympathetic stress reaction, noise consisting of speech also activates the parasympathetic branch leading to a more stable balance of the ANS typically represented by an increased HRV [20] and decreased HR [13]. Umemura & Honda [21] found that listening to classical music suppressed a sympathetic stress reaction (i.e. by keeping a stable ANS balance) compared with rock music or noise—both of which increased activity in the sympathetic control (measured by Mayer wave-related sinus arrhythmia) while simultaneously decreasing activity in the parasympathetic control (measured by respiratory sinus arrhythmia). Moreover, the degree of suppression was positively correlated with subjective reports of comfort [21]. On top of this, individual differences in HR responses to unpleasant noise (high-speed dental engine) have been explained by individuals' familiarity and experience with the acoustic stimuli [22], indicating adaptation to commonly occurring noise sources in daily life. Regarding active task-related listening, elevated stress levels are typically associated with listening in adverse conditions such as noise, which degrade speech recognition and facilitate recruitment of executive cognitive resources [23]. This was illustrated by Holube *et al.* [24], who found stress as measured with electrodermal activity correlated significantly with subjective ratings of listening effort and reported stress for listening to a speech at different signal-to-noise ratios at a constant sound intensity (55 dB), indicating that stress was induced by the effort of listening. In addition to this, the reward or success importance of a given listening task has been found to modulate cardiovascular activity by sympathetic arousal [25]. These findings are in line with the framework for understanding effortful listening (FUEL), which dictates that the effort expended during active listening is modulated by task demand (i.e. listening condition) and personal motivation for engagement [26]. This again highlights the need for considering sound dimensions beyond intensity when assessing the effects of the everyday acoustic environment on the cardiovascular system.

Measurements of the association between human ANS balance and noise intensity have typically been performed under highly controlled conditions (i.e. in the laboratory). This eliminates the potential impact of daily life contexts, which as pointed out above, are expected to play a significant role in ANS reactions. For example, individual sensitivity towards noise might fluctuate with time-of-day, location (e.g. home versus at work) or personal motivation, thus limiting the generalizability of effects measured in the laboratory [26,27]. Notably, in a recent real-world study [28], short-term increases in sound intensity over a 7-day period were associated with a concomitant increase in HR and HRV parameters, but a delayed decrease in the overall HRV reflecting the withdrawal of parasympathetic and elevation of sympathetic control [20], and confirming the role of real-world noise intensity in disrupting ANS stability. In addition, the strength of the association between sound intensity and change in HR was significantly moderated by place/mobility contexts. This highlights the importance of conducting real-world studies examining the effects of everyday sound immersion on the human cardiovascular system [28,29]. This notion has long been acknowledged in city planning [30,31] for example, but has not yet been applied in the study of human stress reactions to everyday sound immersion.

In this paper, we use real-world longitudinal and observational data from hearing aid users to investigate how everyday ANS dynamics are associated with acoustic characteristics of the sound environment. First, we describe the typical daily sound environments encountered by the participants, and second, we model how short-term changes in acoustic characteristics are associated with changes in HR throughout the day. We expand upon the current scientific evidence of how noise affects cardiovascular stress by including dimensions of the acoustic environment related to real-world listening experiences. Data were obtained from the participants' own hearing aids and wearables during daily life.

The sound environment is described using four parameters: sound pressure level (SPL), sound modulation level (SML), signal-to-noise ratio (SNR) and soundscape class, representing distinct characteristics of the momentary sound immersion. SPLs represent the sound intensity and are the most commonly used indicator of the sound wave strength. SPL correlates well with human perception of loudness [32]. SMLs represent temporal amplitude modulation—that is, the degree by which the sound wave amplitude oscillates over short periods of time as found for speech and music [33]. Thus, SML represents the short-term dynamics of the sound wave. SNRs represent a spectral dimension of the sound by differentiating between the level of background noise relative to the level of the signal in decibels. A more positive value indicates less noise relative to the signal. Finally, soundscapes are a qualitative dimension of the acoustic environment assumed to relate to how effortful it is to listen to speech-like sources in the presence of different levels of background noise [34,35].

# 2. Methods

## 2.1. Participants and ethics

Participants were users of Oticon Opn hearing aids (Oticon A/S, Smørum, Denmark), who had signed up for the HearingFitness™ feature via the Oticon ON™ remote control app. When signing up for HearingFitness™ participants submitted their consent agreeing to use of their anonymized data (i.e. no personal identifiers were available) for research purposes on aggregated levels (i.e. no single-case investigation are performed) and agreed that data could be stored on Oticon A/S-owned secure servers. In addition, participants gave specific consent permitting access to their GPS location (as a relative measure in metres between each sample) and health data from Apple HealthKit. All data collection and storage were conducted in a 'privacy by design' manner in accordance with the General Data Protection Regulation (EU regulation 2016/679). No ethical approval was necessary for this study according to Danish National Scientific Ethical Committee (https://www.nvk.dk/forsker/naar-du-anmelder/hvilke-projekter-skal-jeg-anmelde).

## 2.2. Data sources and apparatus

Data were obtained from the commercially available HearingFitness™ feature [36], which is offered to users of Oticon A/S Internet-connected hearing aids. All participants had paired their Oticon Opn™ hearing aids with the Oticon ON™ iOS smartphone app for remote control with HearingFitness™ enabled. The HearingFitness™ program activates automatic logging of sound data from the hearing aid microphones together with extraction of health data from the Apple HealthKit app associated

with a user-owned wearable. No information regarding the type and model of the user-owned wearable was available; however, health data from consumer wearables represent a promising source of data for observational studies [37,38].

We extracted a convenience sample of real-world data from 98 in-market users between June and December 2019. Given the real-world nature of the data and the privacy-preserved extraction, no personal information characterizing the participants (e.g. age, gender) are available. However, the participants represent a random sample of typical hearing aid users, thus, we expect that 6 in 10 are male, and are aged around 74 years [39].

## 2.3. Variables collected

### 2.3.1. Sound data

Sound data concerning the ambient acoustic environment were logged by the smartphone-connected hearing aids. Data consist of three continuous acoustic variables and one discrete soundscape variable. The continuous data represent acoustic characteristics of the momentary sound wave sensed by calibrated hearing aid microphones at ear level. These variables are SPL, SML and SNR, all estimated in a broadband frequency range (0–10 kHz) in decibel units. SPL is the level output estimate from a low-pass infinite impulse-response filter with a time constant of 63 ms. SML is then derived as the difference between a top and bottom tracker (peak and valley detector) of the SPL. The bottom tracker is implemented with a slow dynamic attack time of 1 to 5 s and a fast release time of 30 ms and the top tracker is implemented with the reverse (figs. 3–10 in [40]). SNR is the difference between the bottom tracker and the immediate SPL. Thus, values of SPL and SNR are dynamically changing on the same time scale whereas the SML changes with a slower time constant of up to 5 s. The discrete soundscape variable classifies the momentary sound environment into four soundscapes by a proprietary hearing aid algorithm using SPL, SML and SNR values. The soundscapes are: 'Quiet', 'Noise', 'Speech' and 'Speech in Noise'. Similar soundscape classification from hearing aids has been reported as a potential source for data mining [41]. Data of the acoustic environment were logged every 60 s together with timestamps indicating when the hearing aids were turned on and connected via Bluetooth to a smartphone.

### 2.3.2. Heart rate data

A 5 min running mean HR was extracted from the Apple HealthKit storage (https://developer.apple.com/documentation/healthkit) approximately every 7 min (observed mean = 7.6 min, s.d. = 2.8 min) when a wearable with a heart rate sensor was connected to the smartphone.

## 2.4. Data selection and filtration

In total, 1 115 332 acoustic environment data logs and 522 715 heart rate logs were obtained, which represents approximately 9000 h of bilateral hearing aid use and 61 000 h of data from wearables, respectively. Only data logged between 06.00 and 24.00 were considered valid to avoid confounds from night-time logs that were probably collected when neither the hearing aids nor wearables were being worn.

Data were examined in two stages. First, in order to maximize the data available, the acoustic environment was described using all observations regardless of temporal overlap with HR logs. Second, for subsequent statistical modelling, the data were pre-processed to ensure full overlap between acoustic variables and HR logs. Pre-processing consisted of selecting time-windows of 5 min prior to each HR log and computing the arithmetic average of each acoustic variable within that window. Thus, the fully overlapping data consists of data records with the value of each acoustic data variable estimated from the same time-window as the running mean HR logs. We excluded data records with HRs that were below the 5th percentile and above the 95th percentile of the group mean HR to avoid potential confounds from low-incident HRs [37]. Exclusion of these data ensured normality of residuals from statistical modelling but did not affect the order of regression coefficients or the statistical significance of the included statistical models. Further, we only included participants for whom there were at least 50 overlapping HR and acoustic data logs. After pre-processing, our data consisted of 25 193 data records from 56 participants and 971 unique participant-days. Figure 3a shows the count of data records separated by time-of-day and weekday, and figure 3b shows the density distributions of each variable in the data records.

The effective sampling period of data records was dependent on the proportion of time that a participant simultaneously wore their hearing aid(s) and a wearable while each was connected to a smartphone. As is typical of hearing aid users, participants wore their hearing aids for different durations throughout the day. This was true both within and across participants. Thus, we estimated the effective sampling frequency (i.e. representativeness of samples across time) by computing the average amount of data records each hour between 6.00 and 24.00 per participant across the period covered in the data. The grand average sampling frequency across all 971 participant-days and hours was 2.11 data records per hour (s.d. = 2.42) with a peak at 19.00 (2.21 data records per hour, s.d. = 1.72).

## 2.5. Adjusting for movement

A subset of the data records contained GPS information associated with each HR measure. This was used to examine the possible confounding effect of movement on the relationship between HRs and acoustic environment data. The movement was estimated within the database and extracted for analysis. Specifically, the distance in metres between two consecutive latitude and longitude coordinates was computed using the haversine method. Using this method, for each pair of subsequent latitude ($\varphi_1$, $\varphi_2$) and longitude ($\lambda_1$, $\lambda_2$) coordinates the distance between them in metres, $d$, is estimated as

$$d = 2r\arcsin\left(\sqrt{\sin^2\left(\frac{\varphi_2 - \varphi_1}{2}\right) + \cos(\varphi_1)\cos(\varphi_2)\sin^2\left(\frac{\lambda_2 - \lambda_1}{2}\right)}\right), \tag{2.1}$$

with $r$ being equal to the radius of the earth. Movement in m s$^{-1}$ was then computed by dividing $d$ with the 1 min time-window between each observation and averaging across the 5 min time-window preceding each HR measure. To ensure that only movement due to physical activity was considered, data records with movement that exceeded cycling speed (10 m s$^{-1}$) was excluded.

## 2.6. Statistical analysis

Due to the unbalanced (unequal samples per participant, day, hour etc.) and hierarchical multi-level nature of the data records, associations between variables were quantified using linear mixed-effect (LME) models. All statistical analysis and visualizations were done in R v. 3.6.1. For the application of mixed-effects models, the 'nlme—Linear and Nonlinear Mixed-Effects Models' package (v. 3.1) was used [43]. Visualizations were done with 'ggplot2' package (v. 3.3.2).

### 2.6.1. Associating sound with heart rate

LME models were applied separately for associating mean HR with either the categorical soundscape or the acoustic data (SPL, SML and SNR). The random effects structure was adjusted for individual differences in baseline HR and sensitivity towards each fixed-effect (i.e. random intercepts and slopes) and adjusted for baseline offsets in HRs due to time-of-day (in hours), and weekday nested within individuals [44]. In separate models, adjustment for the movement was applied by adding an additional nested random effect equal to the estimated movement (see the previous section) quantified into 10 equal-sized bins (aka deciles).

### 2.6.2. Testing the moderating effect of soundscape

An interaction model was applied to test the hypothesis that soundscape moderates the association between acoustic data and mean HR. That is, we expect associations between acoustic characteristics and HR to be moderated by listening condition, which is represented by the soundscape data. The interaction model was applied with the same random effects structure as the simpler models testing for main effects. Note that the movement adjustment was not applied for the interaction model due to convergence issues in the model fitting procedure.

### 2.6.3. Model diagnostics

Model diagnostics were conducted in accordance with recommendations in the literature [45], and models adhered to assumptions of normality of residuals and homogeneity of variance among random effect groups (see the electronic supplementary material). The autocorrelation of residuals (i.e. HR observations are not independent) was addressed with a first-order autoregressive structure,

which has previously been shown to improve the goodness-of-fit with similar data of mean HRs [28]. To assess the degree of multicollinearity within the models, particularly with the addition of acoustic characteristics and their interaction with soundscape, the generalized variance inflation factor (GVIF) was computed using the 'car' package v. 3.0.8 [46]. The GVIF is a generalization of the variance inflation factor (VIF) that can be applied to categorical explanatory variables [47]. Values of GVIF less than 4 are usually considered to be acceptable [48].

### 2.6.4. Effect size estimation

Besides inspecting the magnitude and confidence intervals of the standardized LME model coefficients, partial variance explained by the fixed effects, $R_P^2$, was estimated based on recommendations for multi-level hierarchical models [49]. Briefly, the sum of residual sum-of-squares across each grouping level of the models including both fixed and random effects were compared with those from intercept-only models with identical random effects. Explained variance of the fixed effects were then computed as the proportional reduction in prediction error [50],

$$R_P^2 = 1 - \frac{\sigma_F^2 + \tau_F^2 + \gamma_F^2}{\sigma_N^2 + \tau_N^2 + \gamma_N^2},$$ (2.2)

where $\sigma_F^2$ represents the level-one (individual) residual sum-of-squares for the full model; $\tau_F^2$ represents the level-two (weekday) residual sum-of-squares for the full model and $\gamma_F^2$ represents the level-three (time-of-day) residual sum-of-squares for the full model. The terms in the denominator represent the residual sum-of–squares for the same three levels but for the intercept-only (null) model. To estimate the explained variance of the full model including random effect terms ($R_F^2$), the denominator in equation (2.2) was replaced with the total sum-of-squares from the observations (i.e. the 'natural' variance).

# 3. Results

## 3.1. Descriptive statistics

Prior to examining associations between ambient sound data and HR, descriptive analyses and comparisons with previously reported data were conducted to assess data validity.

### 3.1.1. Soundscape and acoustic data

The grand median SPL across all participants was 54.42 dB (s.d. = 6.68 dB), which corresponds to a level just below normal conversational speech (approx. 60 dB). The values of each acoustic characteristic varied according to the soundscape as classified by the hearing aid (figure 1). Specifically, as expected, 'Speech' environments consisted of higher values of raw SNR and SML, while 'Speech in Noise' and 'Noise' had the highest SPL.

We also examined the variation in sound data as a function of time of day. Figure 2a–c shows quartiles of each acoustic characteristic (grand median across participants), while figure 2d shows the relative occurrence of each soundscape by the time of day. As would be expected, the SPL is lowest late in the evening (22.00 and 23.00), slightly higher in the morning (6.00 to 8.00), and highest around lunchtime (12.00) and dinner time (18.00). The proportion of time the sound environment was classified as being 'Quiet' was also highest in the evening (from 20.00) and early morning (6.00 to 9.00). Conversely, the proportion of time the ambient acoustic environment was classified as 'Speech' was greatest early evening (16.00 to 21.00), which corresponds in time to when the SML was also greatest. Similar findings have been reported in other studies [51].

We interpret the temporally fluctuating acoustic environment characteristics as reflecting differing everyday contexts, with figure 2d suggesting that for approximately 65–70% of the day, the acoustic environment is classified as 'Quiet' or 'Speech'. This is similar to the findings of Humes et al. [51], who found that older adults with hearing aids typically spent 60% of their time in 'quiet' or 'speech-only' conditions, while around 10% of the time is being spent in 'pure noise'. This, combined with the face-validity of the data, provide evidence that our acoustic data reflect real-world characteristics.

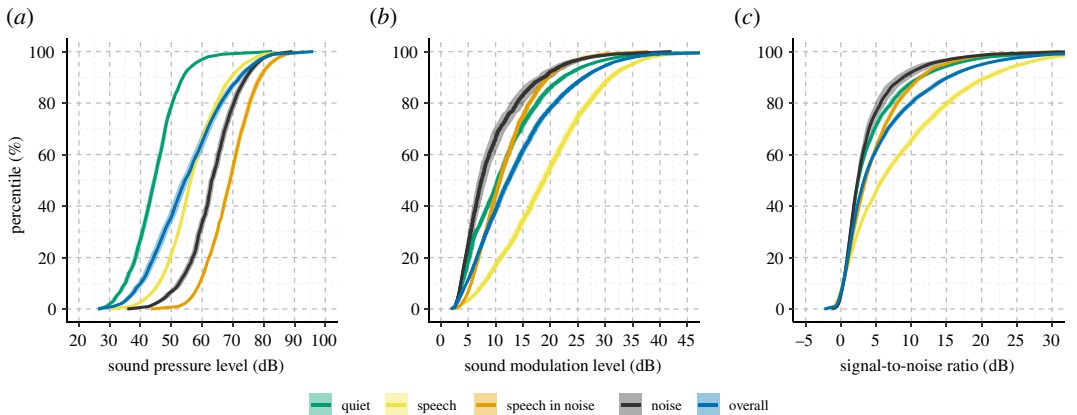

**Figure 1.** Cumulative distribution functions of acoustic data separated by soundscape. Median sound pressure level (*a*), sound modulation level (*b*) and signal-to-noise ratio (*c*) for each percentile across participants. Shaded area represents the 95% confidence interval.

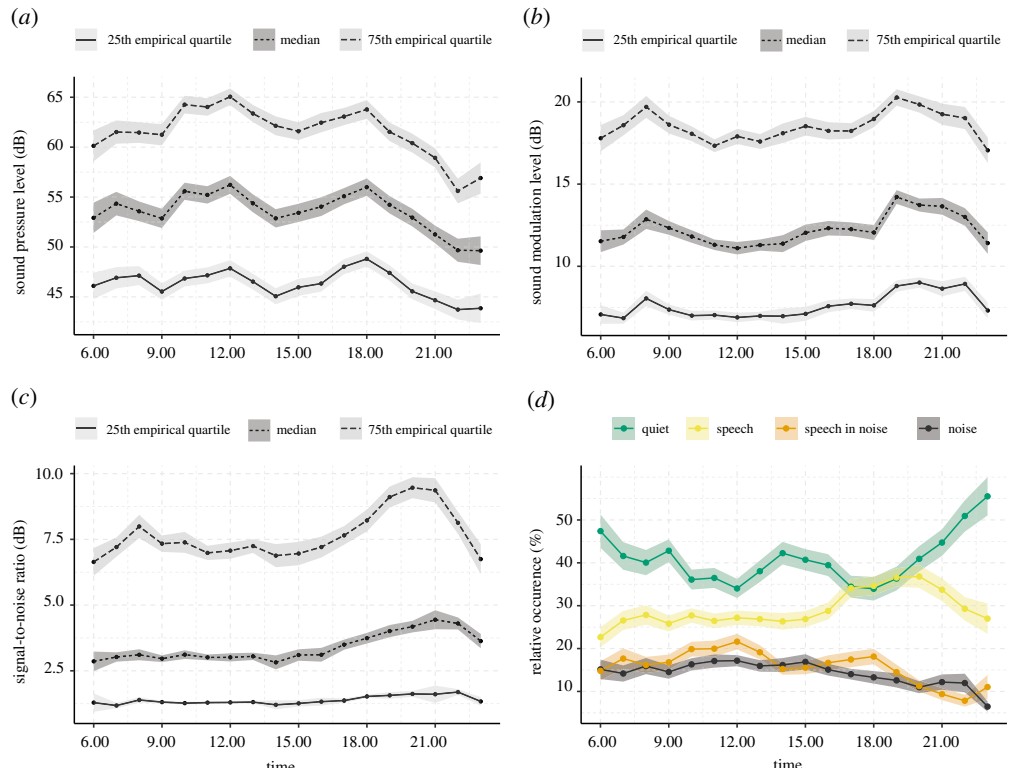

**Figure 2.** Everyday acoustic environment. (*a*–*c*) quartiles of the continuous acoustic data for each hour of the day computed as the grand median across all participants (solid, dashed and dotted lines). (*d*) Relative occurrence of each soundscape for each hour of the day computed as the mean percentage across all participants. Shaded area represents the standard error.

### 3.1.2. Heart rate data

The grand mean HR was 75.60 bpm (s.d. = 6.89 bpm), which is lower than the normative value of 79.1 bpm (s.d. = 14.5 bpm) for individuals aged 18 years and older [52]. Our values might differ due to the age distribution of our population combined with the fact that HR declines with age [53]. Our study population consists of hearing aid users and the average age of first-time hearing aid users is around 69 years [39]. When comparing our data with the mean real-world HR for people aged 71 to 80 years, the values are much closer: 74.2 bpm (s.d. = 11.1 bpm) versus 75.60 bpm (s.d. = 6.89 bpm) here.

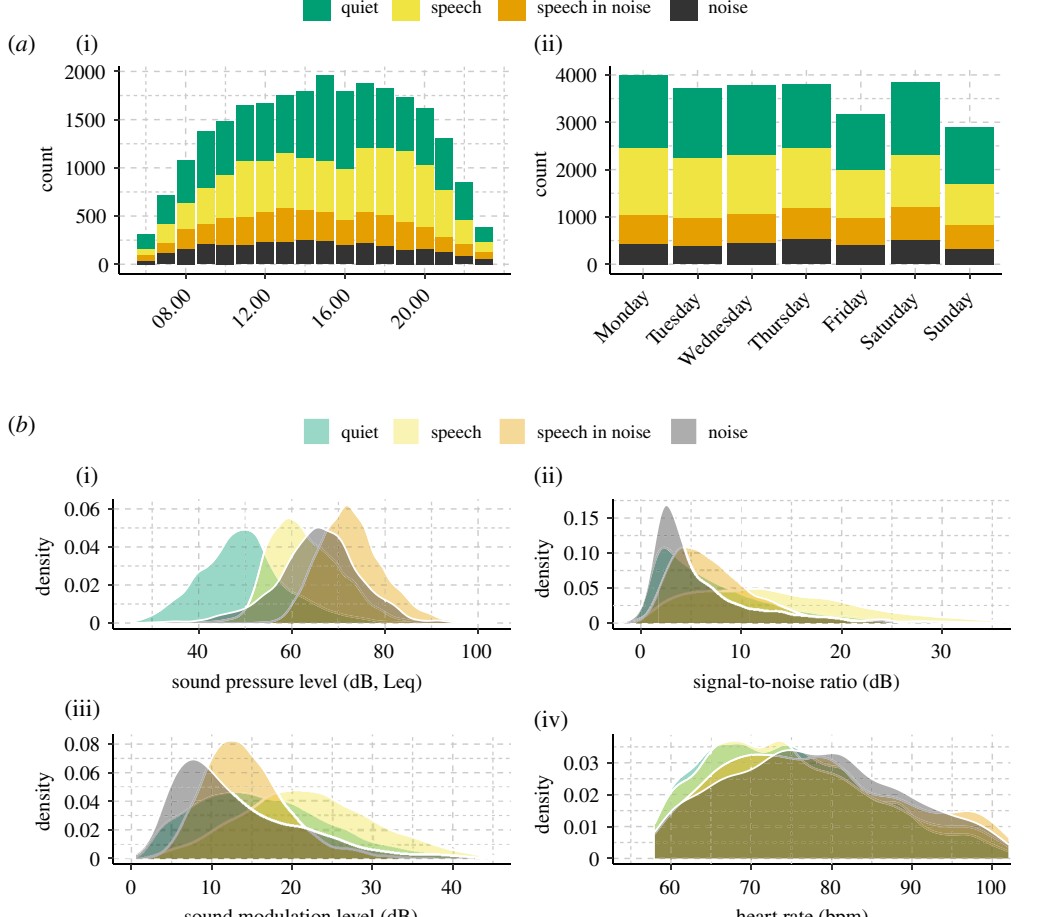

**Figure 3.** Overview of data records for statistically associating mean HR and ambient sound. (*a*) Total counts of data records for each hour of the day (i) and weekday (ii) for each soundscape class (colours). (*b*) Density distributions for each acoustic data variable and the heart rates. Note that data records with HR below the 5th or above the 95th percentile were excluded prior to visualizing (see text for details).

## 3.2. Association between ambient sound and heart rate

The association between ambient sound characteristics and mean HRs was investigated with the subset of data with overlapping acoustic and HR information. We first show descriptive statistics and then the results of LME modelling to formally associate changes in ambient sound with changes in mean HR.

Figure 3 shows summary distributions of the temporally overlapping data records of HR and sound data. Figure 3*a* indicates a close to uniform sampling of observations from 10.00 to 20.00 and across weekdays except for Friday and Sunday, which exhibits approximately 500–1000 less records than the rest of the week. When averaged across all participants, 35% (s.d. = 20%) of data records were registered as being a 'Quiet' soundscape, 32% (s.d. = 18%) as a 'Speech' soundscape, 18% (s.d. = 14%) as a 'Speech in Noise' soundscape, and 15% (s.d. = 12%) as a 'Noise' soundscape.

Density distributions of the continuous acoustic parameters associated with each data record are shown in figure 3*b*(i, ii, iii). Not surprisingly, the sound intensity (SPL) was highest for 'Speech in Noise' and 'Noise' soundscapes with median SPL Leq being 72.47 dB (s.d. = 3.87 dB) and 65.97 dB (s.d. = 7.34 dB), respectively. For comparison, the same value in 'Quiet' was 48.79 dB (s.d. = 5.25 dB). These values correspond well with those reported by El Aarbaoui & Chaix [28] for differing contextual locations (i.e. 'Public' and 'Transportation' versus 'Home'). Regarding SNR, as would be expected, the 'Quiet' and 'Noise' soundscapes exhibited lower median values (Quiet: 4.89 dB, s.d. = 3.86 dB, Noise = 3.81 dB, s.d. = 2.50 dB) compared with 'Speech' (12.50 dB, s.d. = 5.14 dB). These values are in line with previously reported real-world data [54,55]. Lastly, median SML was highest for 'Speech' and 'Quiet' (Speech = 21.77 dB, s.d. = 6.45 dB; Quiet = 15.28 dB, s.d. = 5.84 dB) and lowest for 'Noise' (11.57 dB, s.d. = 4.69 dB), indicating that, indeed, SML represents modulated sound with low levels of noise.

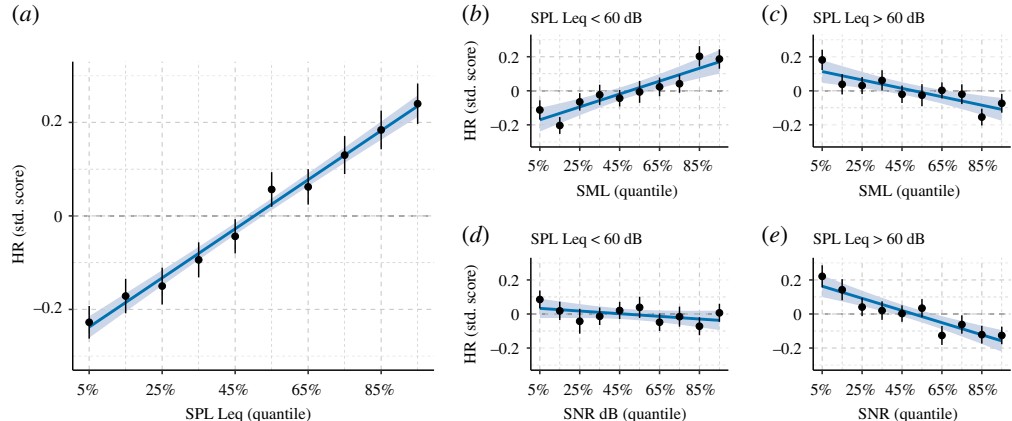

**Figure 4.** Marginal mean heart rate (HR) grouped in non-overlapping decile bins (bin-centres on $x$-axis) of the acoustic characteristics. Solid lines represent best fitting linear regression with 95% CI for the prediction (shaded area). (*a*) HR versus SPL Leq ($\beta = 0.05$, $F_{1,8} = 886.70$, $p < 0.001$, $R^2 = 0.99$). (*b*) HR versus SML at low intensities ($\beta = -0.01$, $F_{1,8} = 2.30$, $p = 0.132$, $R^2 = 0.27$). (*c*) HR versus SML at high intensities ($\beta = -0.04$, $F_{1,8} = 61.86$, $p < 0.001$, $R^2 = 0.86$). (*d*) HR versus SNR at low intensities ($\beta = 0.04$, $F_{1,8} = 45.39.31$, $p < 0.001$, $R^2 = 0.85$). (*e*) HR versus SNR at high intensities ($\beta = -0.02$, $F_{1,8} = 20.76$, $p = 0.002$, $R^2 = 0.72$). See text for details about computing the marginal means.

Figure 3*b*(iv) shows the distribution of HR across all records. Mean HR differed by soundscape with values for 'Quiet': 74.6 bpm (s.d. = 6.9 bpm); 'Speech': 75.9 bpm (s.d. = 7.0 bpm); 'Speech in Noise': 78.2 bpm (s.d. = 8.6 bpm); and 'Noise': 77.6 bpm (s.d. = 6.9 bpm). A repeated-measures ANOVA shows a significant main effect of soundscape ($F(1.94,103.03) = 9.39$, $p = 0.034$), with *post hoc* comparisons using Bonferroni correction indicating that the mean HR for 'Quiet' was significantly lower than for 'Speech' ($p = 0.017$), 'Speech in Noise' ($p < 0.001$), and 'Noise' ($p < 0.001$); the mean HR for 'Speech' was significantly lower than for 'Speech in Noise' ($p = 0.019$), but not for 'Noise' ($p = 0.12$). HR for 'Speech in Noise' and 'Noise' did not significantly differ.

Marginal mean (i.e. across factors) HR at increasing levels of each acoustic characteristic are shown in figure 4. The marginal mean HR was computed by first standardizing each participant's HR and acoustic data (centring and scaling) and then computing the pooled average HR within non-overlapping bins for deciles of SPL, SNR and SML. For example, the first bin in figure 4 (at the 5% quantile on the x-axis) represents the average standardized HR for values of acoustic data falling between the 0% and 10% quantile. The standardization of acoustic characteristics was done to prevent confounds from inter-individual differences (e.g. hearing aid microphone placement and offset) when computing the marginal mean HR.

Since high levels of SNR and SML can occur for both low and high levels of SPL (figure 5), we computed the marginal mean HR versus SML and SNR in two regions of SPL: SPLs below and above the overall median (i.e. median Leq = 60 dB SPL).

The marginal means and fitted linear trends in figure 4 indicate a strong association between acoustic characteristics and HRs. In addition, figure 4*b–e* reveals that the association between derived moments of the acoustic signal (i.e. SNR and SML) and mean HR is conditional on intensity. That is, a comparison of figure 4*d,e* show that SNR is more strongly associated with HR when the SPL Leq is above rather than below 60 dB (i.e. steeper slope of the regression) while figure 4*b,c* indicate that SML is either positively associated with HR at lower SPLs or negatively associated with HR at higher SPLs.

### 3.2.1. Statistical modelling

As stated in the introduction, everyday sounds are not only modulated by intensity but also by dynamic characteristics, perceptual quality (i.e. noisy or clean signals) and type (e.g. conversation, music, traffic). These distinct traits can be approximated by SPL, SML, SNR and soundscape, respectively. However, these acoustic dimensions inherently correlate. For example, high-intensity speech signals (high SPLs) also have a high SML, and if not masked by noise, a high SNR. This is evident from figure 5, which shows the interrelations between SPL, SML, SNR and soundscape as two-dimensional density distributions. The potential confound of multicollinearity in the LME model testing for main effects of acoustic data was assessed by computing the generalized variance inflation factor (GVIF, see

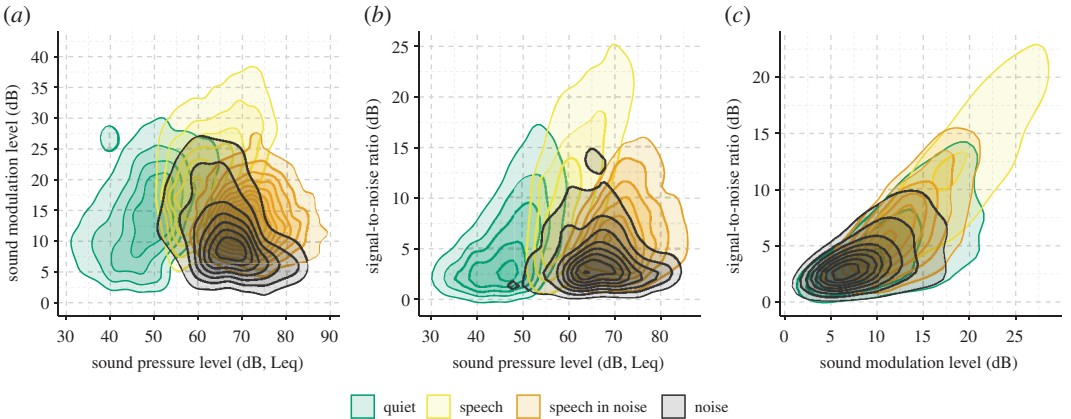

**Figure 5.** Relationship between each acoustic variable and the soundscape in the data records as two-dimensional density distributions. (*a*) SML versus SPL. (*b*) SNR versus SPL. (*c*) SNR versus SML.

**Table 1.** Regression coefficients (β) and 95% confidence intervals of the change in mean heart rate (bpm) associated with a change in soundscape (soundscape model) or a 1 s.d. change in either SPL, SML or SNR (acoustic data model). Models were fitted to either all data records (*n* = 25 193) or to only those records that contained movement data (*n* = 5613). In the latter case, movement was included as a nested random effects term in the adjusted model and left out in the non-adjusted model. Note that regression coefficients are considered significant as long as the 95% confidence interval does not cross zero [42].

| | all data records | | data records with movement | | | |
| --- | --- | --- | --- | --- | --- | --- |
| | non-adjusted | | adjusted | | non-adjusted | |
| | β | 95% CI | β | 95% CI | β | 95% CI |
| soundscape model | | | | | | |
| S versus Q | +1.07 | [+0.81 to +1.33] | +1.46 | [+0.91 to +2.00] | +1.49 | [+0.96 to +2.02] |
| SN versus Q | +2.23 | [+1.89 to +2.57] | +1.85 | [+1.17 to +2.53] | +2.00 | [+1.33 to +2.65] |
| N versus Q | +2.61 | [+2.25 to +2.97] | +1.84 | [+1.03 to +2.66] | +1.96 | [+1.17 to +2.75] |
| acoustic data model | | | | | | |
| SPL | +1.47 | [+1.29 to +1.76] | +1.39 | [+0.89 to +1.90] | +1.44 | [+0.95 to +1.93] |
| SML | +0.72 | [+0.49 to +0.95] | +0.53 | [+0.06 to +0.99] | +0.53 | [+0.09 to +0.98] |
| SNR | −1.03 | [−1.26 to −0.79] | −0.86 | [−1.21 to −0.51] | −0.88 | [−1.21 to −0.55] |

Note: Q, quiet; S, speech; SN, speech in noise; N, noise.

Methods). The GVIF was 1.05, 1.17 and 1.22 for SPL, SML and SNR, respectively, indicating that multicollinearity was not a problem [48]. However, given the high degree of clustering of the continuous acoustic data with soundscape (figure 5), we included the two types of data as independent variables in two separate LME models to predict mean HR. We next fitted the models to a subset of the observations which included GPS data to investigate the influence of controlling for movement activity (adjusted model in table 1) estimated from relative GPS coordinates (see Methods).

Inclusion of sound data to predict heart rates significantly improved model goodness-of-fit when compared with intercept-only models with likelihood-ratio tests (acoustic data: $\chi^2(6) = 723$, $p < 0.001$; soundscape data: $\chi^2(3) = 277$, $p < 0.001$), confirming a significant association between changes in mean HR and the acoustic environment. The estimated partial variance explained was 4.25% for the acoustic data model and 1.43% for the soundscape model (table 2).

The acoustic data model revealed distinct associations between HR and the acoustic characteristics. Specifically, the regression coefficient for SPL is larger than for SML and SNR (non-overlapping confidence intervals), and the regression coefficient for SNR has negative signage, whereas both SPL and SML are positively associated with HR. The movement-adjusted acoustic data model explained approximately 10%-point additional variance (table 2); however, the regression coefficients did not differ between the adjusted and non-adjusted model (table 1).

**Table 2.** Explained variance by fixed effects ($R_P^2$) and the full model including both fixed and random effects ($R_F^2$). Note that data in the interaction model were collapsed across 'speech in noise' and 'noise' soundscapes.

| | $R_P^2$ | $R_F^2$ |
|---|---|---|
| **all data records ($n = 25\ 715$)** | | |
| soundscape model | 1.43% | 56.28% |
| acoustic data model | 4.25% | 57.22% |
| interaction model | 4.54% | 57.30% |
| **data records with movement ($n = 5919$)** | | |
| soundscape model | | |
| *adjusted for movement* | 0.61% | 72.12% |
| *non-adjusted for movement* | 0.78% | 62.11% |
| acoustic data model | | |
| *adjusted for movement* | 2.52% | 72.56% |
| *non-adjusted for movement* | 2.76% | 62.86% |

The regression coefficients for the soundscape model confirmed the trend present in figure 3b(iv) that HR increases as the complexity of the soundscape increases from 'Quiet', to 'Speech', to 'Speech in Noise' and finally 'Noise'. Note that complexity is defined as the interaction between SNR and SPL. High complexity is assigned to soundscapes with low SNR and high SPL (i.e. 'Noise', figure 5). Thus, despite 'Speech in Noise' having the highest SPL (figure 1a), HRs were overall higher in soundscapes classified as noise. The movement-adjusted soundscape model likewise yielded a 10%-point increase in explained variance (table 2), and the coefficients (table 1) suggest that a change from 'Quiet' to 'Speech in Noise' and from 'Quiet' to 'Noise' results in approximately the same change in mean HR. However, these changes are lower in magnitude as was the case for the non-adjusted model. This indicates that, indeed, the movement might have influenced the change in mean HR when contrasting 'Speech' with more complex (i.e. noisier) 'Speech in Noise' and 'Noise' soundscapes.

The median movement in the time-window preceding each HR observation was $0.15\ \mathrm{m\,s^{-1}}$ (s.d. = $1.55\ \mathrm{m\,s^{-1}}$), suggesting that most of these data records were associated with little movement although a significant linear relationship was observed between movement and mean HR (LME adjusted for the participant, $\beta = 0.39$, 95% CI = [+0.22 to +0.57], $t = 4.40$, $p < 0.001$).

## 3.3. Moderating effect of soundscape

The data in table 1 suggest that changes in mean HR throughout the day are significantly associated with changes in the soundscape, and that overall mean HRs are lower in less complex soundscapes (e.g. 'Quiet' versus 'Speech in Noise'). Here, we investigated the extent to which the soundscape moderated the strength of the association between HR and its acoustic characteristics. That is, were participants more sensitive towards acoustic characteristics in certain soundscapes? We would expect this to be true since differences among soundscapes proxy differences in listening conditions. For instance, it is less effortful to understand speech in a quiet environment than in a noisy one. To examine this, we fitted an interaction model for data from 'Quiet', 'Speech' and 'Noisy' soundscapes, reflecting increasing levels of soundscape complexity. The 'Noisy' consisted of data from the 'Speech in Noise' and 'Noise' soundscapes combined since they were highly overlapping in terms of acoustic characteristics (figure 5) and contained the fewest observations (figure 3).

We fitted an interaction model with each acoustic parameter being allowed to interact with soundscape while keeping the same random effects variance structure as before. For this model, we used all data records and did not include movement as a random effect term since the additional degrees-of-freedom would otherwise render the model unidentifiable. The interaction model produced better predictions than both the soundscape model ($\Delta$AIC = 270.5, $\chi^2(11) = 292.54$, $p < 0.001$) and the acoustic data model ($\Delta$AIC = 12.6, $\chi^2(8) = 28.6$, $p < 0.001$) and were able to explain slightly more of the HR variance (table 2). Despite the potentially high covariance between soundscape and acoustic data

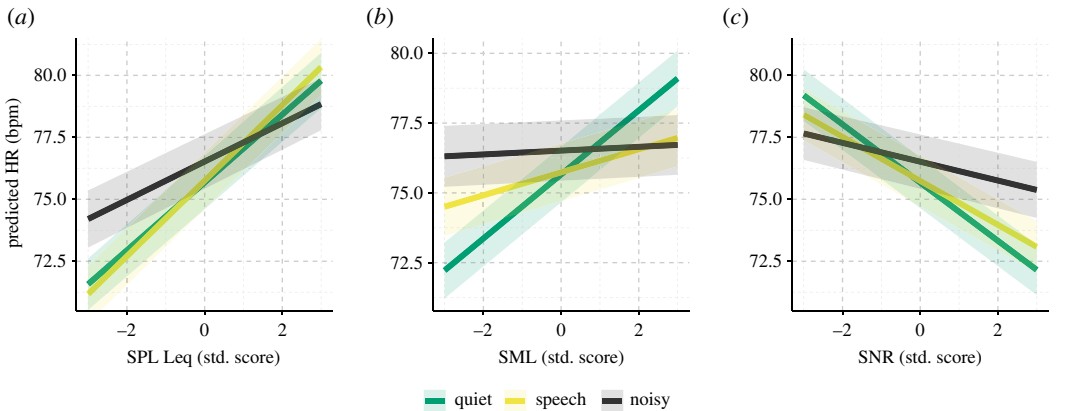

**Figure 6.** Predicted regression lines from the LME interaction model of the coefficients SPL (*a*), SML (*b*) and SNR (*c*). Shaded area represents the standard error of prediction.

(figure 5), the largest GVIF was estimated to be 1.79 (the interaction between soundscape and SNR), which again indicates that multicollinearity was not an issue [48].

The main effects replicated the outcomes of fitting the acoustic data and soundscape models separately (table 1). That is, HRs are higher for more complex soundscapes ($F(2) = 26.39$, $p < 0.001$), more intense and modulated sound environments (SPL: $F(1) = 68.48$, $p < 0.001$; SML: $F(1) = 30.04$, $p < 0.001$), and environments with more background noise (SNR: $F(1) = 15.35$, $p < 0.001$). Additionally, interactions between soundscape and SPL, SML were significant ($F(2) = 5.21$, $p = 0.006$; $F(2) = 15.64$, $p < 0.001$).

Figure 6 shows each level of the three interactions between acoustic variables and soundscape. It indicates that interactions were driven by differences in the strength of association between 'Quiet'/ 'Speech' and 'Noisy' soundscapes. Thus, observed HRs were more strongly associated with sound intensities and modulation levels when the ambient acoustic environment was classified as being favourable for listening (i.e. in quiet and speech-dominated soundscapes).

We assessed the significance of each pairwise interaction by contrasting each coefficient (i.e. slope) with the 'Quiet' soundscape as baseline. For SPLs, the slope for 'Noisy' soundscape was significantly lower than for 'Quiet' ($\beta = -0.60$, s.e. $= 0.23$, $p = 0.008$) whereas the slope for 'Speech' did not differ from the slope for 'Quiet' ($\beta = 0.16$, s.e. $= 0.24$, $p = 0.51$). For SNRs, the slope for 'Noisy' was significantly less steep than for 'Quiet' ($\beta = 0.79$, s.e. $= 0.24$, $p < 0.001$). Finally, for SMLs, slopes for both 'Speech' and 'Noisy' were significantly lower than for 'Quiet' ($\beta = -0.74$, s.e. $= 0.18$, $p < 0.001$; $\beta = -1.08$, s.e. $= 0.21$, $p < 0.001$).

## 3.4. Effect size considerations

The explained variance of the acoustic data model (table 2) suggests that ambient acoustics (i.e. SPL, SML and SNR) explain around 4% of the within-individual variation in 5 min mean HRs. In order to assess the magnitude of the regression coefficients, and to compare our findings from hearing-impaired individuals with those from El Aarbaoui & Chaix [28] using normal-hearing individuals, we re-fitted the acoustic data model after log-transforming the HRs while keeping predictors (SPL, SML and SNR) in their original dB scale. Back-transformation of the coefficients, $\beta$, with $(\exp(\beta) - 1) * 100$ yields percentage change in 5 min mean HR with 1 dB change in the level of either SPL, SML or SNR. After re-fitting, the coefficients were SPL: $\beta = 0.154\%$, 95% CI [+0.127 to +0.181]; SML: $\beta = 0.112\%$, 95% CI $= [+0.078$ to +0.145]; and SNR: $\beta = -0.169\%$, 95% CI $= [-0.209$ to $-0.130]$. For comparison, El Aarbaoui & Chaix [28] documented a 0.141% (95% CI $= [+0.135$ to +0.148]) change in mean HR (5 min window preceding each HR measure) from a 1 dBA change in SPL.

## 4. Discussion

Using real-world data from in-market hearing aids, we were able to investigate the association between exposure to everyday acoustic characteristics and short-term (i.e. within 5 min) changes in HR. The data reflect hearing aid use over several weeks and as such represent the acoustic environment expected from

everyday-life activities (figures 1 and 2), with characteristics of intensity and soundscape occurrences that agree with previous research [1,51,55].

We found that characteristics of the sound environment were significantly associated with changes in heart rate. Specifically, higher SPLs and SMLs were associated with increased HRs, whereas more favourable (higher) SNRs were associated with lowered HR. These results reproduce earlier findings from other research groups [28] and expand the evidence to also include other dimensions (i.e. SML, SNR and soundscape) of the acoustic environment. These effects were not caused by movement (i.e. sound and HR changing in response to latent physical activity). While the movement-adjusted model explained approximately 10% additional HR variance (table 2), the adjusted versus non-adjusted for movement acoustic data models had similar coefficient magnitudes and overlapping confidence intervals (table 1). Thus, the additional explained variance by the adjusted model did not covary with acoustic data—suggesting that the movement predictor captured HR variance unrelated to variance captured by the acoustic data predictors.

The documented associations between acoustic data and HR were stronger in simple listening soundscapes ('Quiet' and 'Speech') compared with soundscapes classified as containing noise, while marginal means revealed that HR moderation by SMLs and SNRs were distinct, depending on sound intensity. Specifically, marginal means suggested that the negative association between SNR and mean HR was most pronounced in sound environments where the SPL Leq exceeded 60 dB, whereas sound modulation was more strongly associated with increases in HR at intensity levels below 60 dB. SPLs are directly associated with loudness perception [32] and have been found to induce activation of the sympathetic branch of the human ANS [5,17,28]. Our study suggests that even everyday SPLs (i.e. levels well below those typically defined as hazardous to the auditory system) also affect heart rate. Thus, we speculate that the ANS is moderated by sound pressure—regardless of the level of sound intensity.

We also documented a positive, albeit smaller, association between HR and SML. SML reflects sound wave modulation. Highly modulated sound is typically characterized by fast oscillating SPLs, which are indicative of speech or music. We speculate that the positive association between SMLs and HRs are, to some extent, caused by conversational task demands [56]. That is, in highly modulated sound environments, listening and speech demands are increased, which leads to slightly increased sympathetic ANS activity. This speculation is corroborated by the fact that the positive association between SML and HR is stronger for low-intensity simpler acoustic environments than for louder noisy environments, i.e. Figures 4d and 6c, which suggests the association is being driven by task-related activities.

We also documented a negative association between real-world HRs and SNRs. This finding corroborates laboratory research within the hearing sciences, which shows that difficult listening conditions (e.g. characterized by low SNRs) increases the listening effort needed for speech understanding, and thus, either elevates the sympathetic stress levels or decreases the parasympathetic ANS activity seen as increased pupil dilation [57,58] or skin conductance [56] and decreased HR variability [59]. This suggests that the ambient SNR, as measured by the hearing aids, could proxy as a real-life indicator of momentary and contextual listening difficulty, while associated changes in HR might indicate the level of mobilized listening effort [26]. However, future studies should include subjective reporting of listening intentions and experiences to reveal specifically the contribution of listening effort and fatigue to changes in everyday ANS activity. For example, experience sampling methods such as ecological momentary assessments for assessing everyday-life listening experiences in hearing aid users [60–62] could be expanded with monitoring of physiological signals.

The sound data used in this study were measured by commercial hearing devices. These devices are typically optimized for low power consumption, which typically would entail lower resolution and accuracy of sound estimators. Despite this, summary statistics of our data agree with other studies using devices specifically developed for research purposes [28]. This highlights a potential for exploiting commercial devices, and in particular, hearing aids, for obtaining real-world and truly ecological evidence about human behavioural and biological reactions to environmental sound stimuli. For example, future studies using a similar set-up could investigate how sensitivity towards specific acoustic characteristics differ between populations with various underlying health pathologies or different degrees of hearing losses [63]. Indeed, previous research has already documented the use of similar hearing aid data logging for policy-making within the hearing or public health domains [64,65].

# 5. Limitations

The participants in our dataset do not represent the general population because all were hearing aid users, thus, they have some degree of hearing loss and are probably older. Some studies suggest that

older people with hearing loss spend more time in quiet sound environments than younger and normal-hearing individuals [66]. Moreover, hearing loss can lead to increased sensitivity towards noisy and loud sound environments, which might have impacted the associations listed in table 1. However, comparing the magnitude of the regression coefficient for SPL with that of El Aarbaoui & Chaix [28] suggests that age and hearing abilities did not affect the associations found here. Further, hearing aids process ambient sound with the goal of improving the SNR by changing how signals are amplified and noise is reduced. Detailed information about these parameters under varying real-world conditions are unavailable in our data. Thus, the effective SNR might differ from the logged ambient SNR presented in figures 1–5. It is thus possible that the findings presented in figure 4e and table 1 would differ among people with normal hearing and/or with unaided impaired hearing. However, as noted in the introduction, laboratory testing of people with unaided hearing impairment also shows negative associations between the SNR of listening tasks and stress reactions measured as electrodermal activity [24].

Heart rates are sensitive to both sympathetic and parasympathetic ANS influence. However, in the current study, only mean HRs are reported. This means that an in-depth investigation of the extent to which stress is caused by an elevation in sympathetic activity or withdrawal of parasympathetic activity is not possible here. Future studies could consider leveraging data available from commercial wearables that measure continuous HRV. This might reveal distinct contributions from the sympathetic and parasympathetic branch of the ANS [20] in response to different sound exposures, representing distinct cognitive and physiological processes. Indeed, short recordings of HRV have been shown to be stable across longitudinal studies [67].

Finally, the data assessed in the current study is considered real-world in nature, which means that there was no control over who used what devices, when and how often. In addition, we did not have information regarding data quality from each wearable device manufacturer and thus cannot give estimates regarding the clinical validity of the HR data. However, we do not expect data quality to have biased our findings since the validity of real-world data has been documented [37], and because the application of filtration during data extraction and application of mixed-effects modelling for statistical analyses minimized the impact of unbalanced observations and inter-individual differences in measures of HR.

# 6. Conclusion

This study is the first to examine the association between real-world human HRs and multidimensional characteristics of the ambient acoustic environment with longitudinal data. Our results indicate that ambient sound intensity is positively associated with heart rates. In addition, we document that the real-world ambient signal-to-noise ratios are associated with lowered HRs, suggesting that sound conditions which reduce the auditory perceptual load and listening effort de-stress the human cardiovascular system [35,57,58,68]. This finding is supported by a documented effect of soundscape on the strength of the association between acoustic characteristics and ANS reactions. That is, in favourable listening conditions, acoustic characteristics have the strongest association with changes in HR.

In summary, our findings suggest a possible mixed influence of everyday sounds on cardiovascular stress, and that the relationship is more complex than is seen from an examination of sound intensity alone. Furthermore, our findings highlight the importance of including exposure to ambient sound in models predicting human physiology and demonstrate that data logging with commercially available devices can be used to study how ecological everyday acoustic environments impact human physiological reactions.

Data accessibility. There are ethical restrictions on publicly sharing the dataset. The consent given by users did not explicitly detail sharing of the data in any format; this limitation is in keeping with EU General Data Protection Regulation and is imposed by the Research Ethics Committees of the Capital Region of Denmark. Data can be obtained by contacting the corresponding author and signing a non-disclosure agreement. Code for conducting the analysis presented in the study can be accessed via the Open Science Framework (doi:10.17605/OSF.IO/RC37Z).
Authors' contributions. J.H.C. wrote the manuscript, analysed the data, produced the figures and came up with the initial hypothesis. G.H.S. critically reviewed and contributed to the manuscript. M.P. collected and made available the data used. N.H.P. critically reviewed the manuscript and supplied technical aspects related to the hearing aid technology.
Competing interests. We declare we have no competing interests.
Funding. The work of J.H.C., G.H.S. and N.H.P. was partly funded from the European Union's Horizon 2020 research and innovation programme under grant agreement no. 727521.

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
