## [Peer Review File · Royal Society Open Science]

Review History

RSOS-201345.R0 (Original submission)

Review form: Reviewer 1

Is the manuscript scientifically sound in its present form?

Yes

Are the interpretations and conclusions justified by the results?

Yes

Is the language acceptable?

Yes

Do you have any ethical concerns with this paper?

Yes

Have you any concerns about statistical analyses in this paper?

No

Recommendation?

Major revision is needed (please make suggestions in comments)

Comments to the Author(s)

This manuscript describes the relationships between Heart Rate (HR) and several acoustical parameters and categories as extracted with a commercial hearing aid.

Although I very much appreciate the attempt to collect real-world data to address the research questions, I have several comments with respect to the methods and description of the methods.

First of all, more information about the methods is required. The authors could provide more details about the device (and data quality) used to collect heart rate data. Also, please provide more background in the introduction about the validity of HR as a measure of listening effort? Or stress? HR is an index of SNS activity but is also influenced by the PNS - please describe the underlying physiological mechanism. Changes in stress do not immediately reflect changes in HR and this relationship also differs between individuals. Please discuss the limitations of the HR measure as collected in the daily-life conditions (i.e, no baseline, no posture correction).

The authors do correct for changes in GPS location at some point, but it is unclear how location information directly translates into movement (as participants could change location without moving (and without much effort) when in a bus or being a passenger in a car, and move /train without changing locations when exercising on their home gym station, for example).

The literature review mainly focuses on the effect of environmental noise on HR. However, the current study aim seems more related to the effect of "high auditory task load" on HR, and not just the effect of any noise per se. Please make a clear distinction between these two aims in the literature review. Please also include more literature relevant to the actual current aim (ie studies on auditory task load using cardiovascular or other physiological biomarkers of effort and stress). As indicated by the current results and as suggested by literature from the same group (Lunner et al.), high noise levels are fairly uncommon in daily life. So the focus of the literature review needs to be adapted accordingly.

Furthermore, more detail is required a priori about the statistical analyses performed. Several analyses are performed, but they are not properly introduced in a statistical methods section. Therefore, the results are difficult to follow, and the connection between research questions and analyses is not always clear. Also, it would be helpful to clearly provide a description of the parameters analysed and the variables controlled for (like the location information).

Ethics: I'm not familiar with Danish law, but I find it remarkable that this study can be performed without informed consent procedure and some form of ethical approval. It would be appropriate to let an ethics committee test and confirm the statement of the authors ("no approval needed"). The authors could then refer to that decision in the ethics statement.

More information is required regarding the selection of users. Were these regular hearing aid users selected from a pool whose data were available to Oticon? Or were they selected via hearing aid dispenser data? Please explain.

The lack of information about the age of the listeners may be problematic in case the influence of auditory load interacts with age. It is known that age influences HR, so this is not unlikely. Please discuss.

Figures: Figures are difficult to read. The position of the labels on the axes is off, and the labels overlap. On greyscale printed paper, there is no visible difference between the colours used to indicate the categories etc. Figure citations are off as well (Figure 55 instead of 5 etc.).

Page 18: Pearson correlations are used to assess the association between sound variables. I wonder whether this approach is the most appropriate - what about calculating ICCs?

Line 349: Speech in noise versus noise differ in a relevant dimension as the motivation of the listener may differ to a great extent between these conditions. Listeners may try to listen to the speech in noise (and may feel stressed when this is difficult or impossible). Alternatively, they may be immersed in a noisy condition without feeling stressed about this as they don't have to perceive any speech anyway (eg. when walking on a train platform). Hence, merging these two categories reduces the sensitivity of the analyses to detect meaningful effects.

Lines 396-401: HR is also influenced by PNS activity. Please adjust this paragraph accordingly.

minor comments:

lines 57-59: please clarify whether the relationship cited was or was not statistically significant. The current phrasing is unclear.

Line 184: a word is missing

Hearing aid classification: This is a black box - no info is provided about the methods. Is there any supporting or background information about the method and validity of the classification algorithm in this user group in daily life conditions?

Relevant literature: Studies performed by Holube et al. - they also assess relationships between acoustical features and subjective effort and stress. von Gablenz, P., Kowalk, U., Bitzer, J., Meis, M., & Holube, I. (2019). Individual hearing aid benefit: Ecological momentary assessment of hearing abilities. In Proceedings of the International Symposium on Auditory and Audiological Research (Vol. 7, pp. 213-220).

Line 92: "were" should be "was"

lines 112-113: please explain this further and provide relevant references for this statement.

Line 158: what happens with the data of the additional 2 mins per 7 min period?

lines 312-313: please use HR consistently (instead of pulse rate)

Line 313: SN should be SNR, I assume

Line 316: were should be was.

Having a native English speaker amongst the co-authors may be exploited a bit more to prevent these small typos and grammatical errors :).

Review form: Reviewer 2

Is the manuscript scientifically sound in its present form?

Yes

Are the interpretations and conclusions justified by the results?

No

Is the language acceptable?

Yes

Do you have any ethical concerns with this paper?

No

Have you any concerns about statistical analyses in this paper?

Yes

Recommendation?

Reject

Comments to the Author(s)

This manuscript left me with mixed feelings. I think that the main research rationale presented in the introduction section – sound characteristics beyond intensity may influence ANS activity and cardiovascular health – is interesting and original. I also liked the well-written introduction section that provides a nice overview of the existing literature and the straightforward description of the methods. Unfortunately, the results and discussion sections do not keep up with the high quality of the first two sections. Moreover, there also seems to be a mismatch between the rationale presented in the introduction and the described methods and results. In sum, I do not think that the manuscript warrants publication in its current form. A thorough revision might bring it above the publication threshold, but the main contribution of the manuscript would be descriptive information. Due to the lack of appropriate measures, it will not significantly advance the understanding of the effect of sound characteristics on ANS and cardiovascular health.

Please let me refer to my main concerns in more detail:

1) There is a mismatch between the employed methods and the research rationale presented in the introduction section. First, to demonstrate that sound characteristics influence cardiovascular health, one would obviously have to assess cardiovascular health. The presented work failed to do so. Second, the introduction nicely points out that it is mainly sympathetic activity that has been associated with poor health outcomes and it also shows that preceding work on this topic has differentiated between sympathetic and parasympathetic responses. Given the differentiated view offered in the introduction, I was surprised to read that the presented work only assessed heart rate as outcome. Heart rate does not allow the separation of sympathetic and parasympathetic effects and does thus constitute a step backwards in relation to preceding work on the topic.

I understand that there might have been practical constraints that prevented the authors from assessing health outcomes and specific indicators of sympathetic and parasympathetic ANS activity, but this does not resolve the fundamental issue: To add to and extend the research discussed in the introduction section, one needs to assess specific indicators of sympathetic (and parasympathetic) activity and collect data related to cardiovascular health.

2) It is difficult to find in the results section the crucial tests that addressed the main research aim. I can understand the wish to provide a comprehensive description of the collected data, but the results section presents so much descriptive information that it is difficult to locate the most important information – the statistical tests that demonstrate the impact of SPL, SML,

SNR, and sound scape on HR. I would recommend that the authors revise the results section to provide more guidance and make more salient what the important information is.

3) The authors' conclusions sometimes seem to be based on visual inspection of graphs or the interpretation of the absolute numbers instead on inferential tests (see for instance the last sentence of the first paragraph on page 17 or the sentence beginning with "The most correlated parameters..." In the first paragraph on page 18). The authors should revise their results and discussion sections to make sure that all conclusions are backed up by appropriate statistical tests.

4) Given the largely exploratory testing strategy, I am concerned about alpha (type-I) error inflation. The authors used alpha error control strategies for some of their post-hoc comparisons, but they did not control for the overall high number of tests that they conducted. I would recommend to either limit p-value based testing to tests that can be justified by a hypothesis or to employ an alpha error control strategy that takes into account the total number of p-value based tests that were conducted.

5) P-value based testing following a Neyman-Pearson approach requires a careful determination of sample size using an a priori power analysis. I would appreciate if the authors could include information on how they determined the sample size of the presented study.

6) I was surprised to read that the authors expected heart rate to show a uniform distribution (second sentence on page 10). It was the first time that I read that all heart rate values should be equally likely. Most papers that I read assumed a normal distribution for heart rate. I also think that some of the statistical procedures that the authors employed to not go well with uniformly distributed data. Moreover, I was wondering whether the authors also had this expectation of uniformly distributed data for the sound parameters that they analysed and whether they applied the same outlier exclusion procedure to these variables.

7) In my pdf version of the manuscript, the readability of some of the figures was low due to overlapping text.

Review form: Reviewer 3

Is the manuscript scientifically sound in its present form?

Yes

Are the interpretations and conclusions justified by the results?

Yes

Is the language acceptable?

Yes

Do you have any ethical concerns with this paper?

No

Have you any concerns about statistical analyses in this paper?

Yes

Recommendation?

Major revision is needed (please make suggestions in comments)

Comments to the Author(s)

My general comments are that I enjoyed the paper but thought: 1) the presentation of the statistical analyses and the reasoning behind them needed to be clearer. This includes variance explained in the regression analyses. Also I saw the ANOVA results for soundscape but did not see the statistics for the raw SPL, SML and SNR. It seems like you meant to include that? 2) More discussion on HR and HRV in the introduction and discussion section is needed. 3) More on why SML is relevant in introduction.

Specific comments:

Introduction

Page 6

Line 89-90 Not sure on sentence saying concomitant increase in HR and HRV values as many HRV values typically decrease with increases in HR

Please add more discussion regarding HRV

Methods

Page 7

Do the wearables provide any additional data than HR?

Page 8

Line 129 Define wearables. Are these all watches?

Line 140 Define SML acronym. Also, why do we care about SML? Obvious why we care about SNR and SPL, not as clear about SML. I assume SML will typically be highest in conditions where there is speech or music and may be relevant to the HA digital noise reduction.

Page 10

Line 175 Do you have another source that used similar exclusion criteria? Why exclude the lowest 5 and upper 5%. Was it effective in removing not only low incident HR but also artifact? Also, what was done if anything to remove artifact?

Page 11

How was movement data specifically used in analyses of HR?

Line 206 What does "nlme" stand for? I assume non-linear and linear mixed effects models

Results

Page 12

Define evening, late at night, early in morning.

Page 13-14

Strange wording related to stating that the results here are consistent with other studies using similarly aged participants. HR does not indicate or determine age (perhaps translation issue here?)

Page 14

Line 248 I thought it was 5:00-24:00 before?

Page 15

Line 261 Figure 3 not 33

Page 16

Having trouble finding statistics indicating the relationship between HR and SNR, SML and SPL aside from those just looking at differences for HR between soundscapes. I thought that was described in methods. Figure 4, I am guessing indicates the relationships but does not give any statistics regarding the relationships

Line 276-280

Need more explanation of what bins and marginal means are and why they were used. What exactly is a bin center quantile in terms of how that relates to raw SPL, SNR and SML?

Line 293 Once again need to tell why we care about SML

Page 18

Line 301 Figure 5, not Figure 55

Not sure the discussion on which factors were more correlated is appropriate or the phrase "most correlated parameters", typically just talk about which were significant.

Page 19

Line 314. Not sure why any of this is important, explain offsets and intercepts.

Page 20

Statistics table seems like it is missing other relevant information above and beyond regression coefficients and Cis. Particularly for the acoustics data model. Such as explained variance, etc. You gave the information that movement, when included explained 32.8% more HR variance, please give values for this for all rows in table with and without movement. I think that seems some of the most important information.

Line 326 Reword sentence, start with: Although adjusted vs. non-adjusted for movement models had similar and overlapping confidence intervals, adjusted models explained an additional 32.8% of HR variance.

Page 21

Line 335-337. Be more precise. You are talking about the regression coefficient size and signage I believe.

Page 22

Line 358 What exactly is a variance inflation factor and how does one interpret 1.79 as being small?

Line 368-369 when the soundscapes were more optimal for listening.....not sure how that is defined

Page 24

More discussion on heart rate literature

Line 402. Does SML simply indicate when speech or music was present? Quiet and noise conditions may at times have neither speech or noise signals, hence lower SML. Listening to speech and music would be cognitively engaging and may increase HR.

Decision letter (RSOS-201345.R0)

Dear Dr Christensen

The Editors assigned to your paper RSOS-201345 "Everyday sound exposure and the association with human heart rate: evidence from real-world data logging" have now received comments from reviewers and would like you to revise the paper in accordance with the reviewer comments and any comments from the Editors. Please note this decision does not guarantee eventual acceptance.

Please submit your revised manuscript and required files (see below) no later than 21 days from today's (ie 19-Oct-2020) date. Note: the ScholarOne system will 'lock' if submission of the revision is attempted 21 or more days after the deadline. If you do not think you will be able to meet this deadline please contact the editorial office immediately.

Best regards,

on behalf of Dr César Lima (Associate Editor) and Pete Smith (Subject Editor)
openscience@royalsociety.org

Reviewer comments to Author:

Reviewer: 1
Comments to the Author(s)

This manuscript describes the relationships between Heart Rate (HR) and several acoustical parameters and categories as extracted with a commercial hearing aid.

Although I very much appreciate the attempt to collect real-world data to address the research questions, I have several comments with respect to the methods and description of the methods.

First of all, more information about the methods is required. The authors could provide more details about the device (and data quality) used to collect heart rate data. Also, please provide more background in the introduction about the validity of HR as a measure of listening effort? Or stress? HR is an index of SNS activity but is also influenced by the PNS - please describe the underlying physiological mechanism. Changes in stress do not immediately reflect changes in HR and this relationship also differs between individuals. Please discuss the limitations of the HR measure as collected in the daily-life conditions (i.e, no baseline, no posture correction).

The authors do correct for changes in GPS location at some point, but it is unclear how location information directly translates into movement (as participants could change location without moving (and without much effort) when in a bus or being a passenger in a car, and move /train without changing locations when exercising on their home gym station, for example).

The literature review mainly focuses on the effect of environmental noise on HR. However, the current study aim seems more related to the effect of "high auditory task load" on HR, and not just the effect of any noise per se. Please make a clear distinction between these two aims in the literature review. Please also include more literature relevant to the actual current aim (ie studies

on auditory task load using cardiovascular or other physiological biomarkers of effort and stress). As indicated by the current results and as suggested by literature from the same group (Lunner et al.), high noise levels are fairly uncommon in daily life. So the focus of the literature review needs to be adapted accordingly.

Furthermore, more detail is required a priori about the statistical analyses performed. Several analyses are performed, but they are not properly introduced in a statistical methods section. Therefore, the results are difficult to follow, and the connection between research questions and analyses is not always clear. Also, it would be helpful to clearly provide a description of the parameters analysed and the variables controlled for (like the location information).

Ethics: I'm not familiar with Danish law, but I find it remarkable that this study can be performed without informed consent procedure and some form of ethical approval. It would be appropriate to let an ethics committee test and confirm the statement of the authors ("no approval needed"). The authors could then refer to that decision in the ethics statement.

More information is required regarding the selection of users. Were these regular hearing aid users selected from a pool whose data were available to Oticon? Or were they selected via hearing aid dispenser data? Please explain.

The lack of information about the age of the listeners may be problematic in case the influence of auditory load interacts with age. It is known that age influences HR, so this is not unlikely. Please discuss.

Figures: Figures are difficult to read. The position of the labels on the axes is off, and the labels overlap. On greyscale printed paper, there is no visible difference between the colours used to indicate the categories etc. Figure citations are off as well (Figure 55 instead of 5 etc.).

Page 18: Pearson correlations are used to assess the association between sound variables. I wonder whether this approach is the most appropriate - what about calculating ICCs?

Line 349: Speech in noise versus noise differ in a relevant dimension as the motivation of the listener may differ to a great extent between these conditions. Listeners may try to listen to the speech in noise (and may feel stressed when this is difficult or impossible). Alternatively, they may be immersed in a noisy condition without feeling stressed about this as they don't have to perceive any speech anyway (eg. when walking on a train platform). Hence, merging these two categories reduces the sensitivity of the analyses to detect meaningful effects.

Lines 396-401: HR is also influenced by PNS activity. Please adjust this paragraph accordingly.

minor comments:

lines 57-59: please clarify whether the relationship cited was or was not statistically significant. The current phrasing is unclear.

Line 184: a word is missing

Hearing aid classification: This is a black box - no info is provided about the methods. Is there any supporting or background information about the method and validity of the classification algorithm in this user group in daily life conditions?

Relevant literature: Studies performed by Holube et al. - they also assess relationships between acoustical features and subjective effort and stress. von Gablenz, P., Kowalk, U., Bitzer, J., Meis,

M., & Holube, I. (2019). Individual hearing aid benefit: Ecological momentary assessment of hearing abilities. In *Proceedings of the International Symposium on Auditory and Audiological Research* (Vol. 7, pp. 213-220).

Line 92: "were" should be "was"

lines 112-113: please explain this further and provide relevant references for this statement.

Line 158: what happens with the data of the additional 2 mins per 7 min period?

lines 312-313: please use HR consistently (instead of pulse rate)

Line 313: SN should be SNR, I assume

Line 316: were should be was.

Having a native English speaker amongst the co-authors may be exploited a bit more to prevent these small typo's and grammatical errors :).

Reviewer: 2

Comments to the Author(s)

This manuscript left me with mixed feelings. I think that the main research rationale presented in the introduction section – sound characteristics beyond intensity may influence ANS activity and cardiovascular health – is interesting and original. I also liked the well-written introduction section that provides a nice overview of the existing literature and the straightforward description of the methods. Unfortunately, the results and discussion sections do not keep up with the high quality of the first two sections. Moreover, there also seems to be a mismatch between the rationale presented in the introduction and the described methods and results. In sum, I do not think that the manuscript warrants publication in its current form. A thorough revision might bring it above the publication threshold, but the main contribution of the manuscript would be descriptive information. Due to the lack of appropriate measures, it will not significantly advance the understanding of the effect of sound characteristics on ANS and cardiovascular health.

Please let me refer to my main concerns in more detail:

1) There is a mismatch between the employed methods and the research rationale presented in the introduction section. First, to demonstrate that sound characteristics influence cardiovascular health, one would obviously have to assess cardiovascular health. The presented work failed to do so. Second, the introduction nicely points out that it is mainly sympathetic activity that has been associated with poor health outcomes and it also shows that preceding work on this topic has differentiated between sympathetic and parasympathetic responses. Given the differentiated view offered in the introduction, I was surprised to read that the presented work only assessed heart rate as outcome. Heart rate does not allow the separation of sympathetic and parasympathetic effects and does thus constitute a step backwards in relation to preceding work on the topic.

I understand that there might have been practical constraints that prevented the authors from assessing health outcomes and specific indicators of sympathetic and parasympathetic ANS activity, but this does not resolve the fundamental issue: To add to and extend the research discussed in the introduction section, one needs to assess specific indicators of sympathetic (and parasympathetic) activity and collect data related to cardiovascular health.

2) It is difficult to find in the results section the crucial tests that addressed the main research aim. I can understand the wish to provide a comprehensive description of the collected data, but the results section presents so much descriptive information that it is difficult to locate the most important information – the statistical tests that demonstrate the impact of SPL, SML, SNR, and sound scape on HR. I would recommend that the authors revise the results section to provide more guidance and make more salient what the important information is.

3) The authors' conclusions sometimes seem to be based on visual inspection of graphs or the interpretation of the absolute numbers instead on inferential tests (see for instance the last sentence of the first paragraph on page 17 or the sentence beginning with "The most correlated parameters..." In the first paragraph on page 18). The authors should revise their results and discussion sections to make sure that all conclusions are backed up by appropriate statistical tests.

4) Given the largely exploratory testing strategy, I am concerned about alpha (type-I) error inflation. The authors used alpha error control strategies for some of their post-hoc comparisons, but they did not control for the overall high number of tests that they conducted. I would recommend to either limit p-value based testing to tests that can be justified by a hypothesis or to employ an alpha error control strategy that takes into account the total number of p-value based tests that were conducted.

5) P-value based testing following a Neyman-Pearson approach requires a careful determination of sample size using an a priori power analysis. I would appreciate if the authors could include information on how they determined the sample size of the presented study.

6) I was surprised to read that the authors expected heart rate to show a uniform distribution (second sentence on page 10). It was the first time that I read that all heart rate values should be equally likely. Most papers that I read assumed a normal distribution for heart rate. I also think that some of the statistical procedures that the authors employed to not go well with uniformly distributed data. Moreover, I was wondering whether the authors also had this expectation of uniformly distributed data for the sound parameters that they analysed and whether they applied the same outlier exclusion procedure to these variables.

7) In my pdf version of the manuscript, the readability of some of the figures was low due to overlapping text.

Reviewer: 3

Comments to the Author(s)

My general comments are that I enjoyed the paper but thought: 1) the presentation of the statistical analyses and the reasoning behind them needed to be clearer. This includes variance explained in the regression analyses. Also I saw the ANOVA results for soundscape but did not see the statistics for the raw SPL, SML and SNR. It seems like you meant to include that? 2) More discussion on HR and HRV in the introduction and discussion section is needed. 3) More on why SML is relevant in introduction.

Specific comments:

Introduction

Page 6

Line 89-90 Not sure on sentence saying concomitant increase in HR and HRV values as many HRV values typically decrease with increases in HR

Please add more discussion regarding HRV

Methods

Page 7

Do the wearables provide any additional data than HR?

Page 8

Line 129 Define wearables. Are these all watches?

Line 140 Define SML acronym. Also, why do we care about SML? Obvious why we care about SNR and SPL, not as clear about SML. I assume SML will typically be highest in conditions where there is speech or music and may be relevant to the HA digital noise reduction.

Page 10

Line 175 Do you have another source that used similar exclusion criteria? Why exclude the lowest 5 and upper 5%. Was it effective in removing not only low incident HR but also artifact? Also, what was done if anything to remove artifact?

Page 11

How was movement data specifically used in analyses of HR?

Line 206 What does "nlme" stand for? I assume non-linear and linear mixed effects models

Results

Page 12

Define evening, late at night, early in morning.

Page 13-14

Strange wording related to stating that the results here are consistent with other studies using similarly aged participants. HR does not indicate or determine age (perhaps translation issue here?)

Page 14

Line 248 I thought it was 5:00-24:00 before?

Page 15

Line 261 Figure 3 not 33

Page 16

Having trouble finding statistics indicating the relationship between HR and SNR, SML and SPL aside from those just looking at differences for HR between soundscapes. I thought that was described in methods. Figure 4, I am guessing indicates the relationships but does not give any statistics regarding the relationships

Line 276-280

Need more explanation of what bins and marginal means are and why they were used. What exactly is a bin center quantile in terms of how that relates to raw SPL, SNR and SML?

Line 293 Once again need to tell why we care about SML

Page 18

Line 301 Figure 5, not Figure 55

Not sure the discussion on which factors were more correlated is appropriate or the phrase "most correlated parameters", typically just talk about which were significant.

Page 19

Line 314. Not sure why any of this is important, explain offsets and intercepts.

Page 20

Statistics table seems like it is missing other relevant information above and beyond regression coefficients and Cis. Particularly for the acoustics data model. Such as explained variance, etc. You gave the information that movement, when included explained 32.8% more HR variance, please give values for this for all rows in table with and without movement. I think that seems some of the most important information.

Line 326 Reword sentence, start with: Although adjusted vs. non-adjusted for movement models had similar and overlapping confidence intervals, adjusted models explained an additional 32.8% of HR variance.

Page 21

Line 335-337. Be more precise. You are talking about the regression coefficient size and signage I believe.

Page 22

Line 358 What exactly is a variance inflation factor and how does one interpret 1.79 as being small?

Line 368-369 when the soundscapes were more optimal for listening.....not sure how that is defined

Page 24

More discussion on heart rate literature

Line 402. Does SML simply indicate when speech or music was present? Quiet and noise conditions may at times have neither speech or noise signals, hence lower SML. Listening to speech and music would be cognitively engaging and may increase HR.

===PREPARING YOUR MANUSCRIPT===

===PREPARING YOUR REVISION IN SCHOLARONE===

Author's Response to Decision Letter for (RSOS-201345.R0)

See Appendix A.

RSOS-201345.R1 (Revision)

Review form: Reviewer 1

Is the manuscript scientifically sound in its present form?

Yes

Are the interpretations and conclusions justified by the results?

Yes

Is the language acceptable?

No

Do you have any ethical concerns with this paper?

No

Have you any concerns about statistical analyses in this paper?

No

Recommendation?

Accept with minor revision (please list in comments)

Comments to the Author(s)

Review of: "The everyday acoustic environment and its association with human heart rate: evidence from real-world data logging with hearing aids and wearables."

Although the readability and clarity of the revised version of this manuscript is improved, I still have several comments regarding the description of the results.

Main comments

page 4-5: Both subjective and objective measures of stress can provide valuable insight into the effects of noise and high task demand on stress. Moreover, especially the combination of such measures is useful (e.g. collecting subjective data in the present study would have been truly interesting as well). I suggest the authors change the conclusion of the first partial paragraph on page 5. Note that I refer to the version of the MS including track changes.

Lines 99-101: RSA is a measure primarily influenced by vagal tone. This is not the same as directly reflecting "sympathetic suppression". Please change the wording of this statement.

Results: still the results section is hard to follow. This is caused by not introducing the sections and analyses properly (in the statistical analysis paragraph: please describe which question will be answered by which analysis). For example, properly introduce section "moderating effect of soundscape". Also, a lot of statements and entire paragraphs include reflections on the (validity of) the results, comparison with previous studies and interpretation of the results. These should be moved to the discussion section. For example, lines 356-357, lines 368-372, line 363, first paragraph of page 19, line 406, line 411, lines 510-etc.

Figure 4 and interpretation of the results. All participants were hearing aid users (Oticon OpN hearing aids). This aspect should be discussed more thoroughly in the paper. It means that *despite* the use of this hearing aid, the results show the associations shown in Figure 4. As this advanced hearing aid will process and alter the sound characteristics, we may assume stronger

associations in linear hearing aids. Furthermore, as the directional and noise reduction features of this hearing aid become more effective at higher sound levels, it seems remarkable that the relationship between SNR and HR is especially observed for higher SPL levels. Does this suggest that these effects of the hearing aid are marginal? Or were these turned off? Please put the results in the context of the hearing aid processing algorithm, as the sound characteristics as analysed in this manuscript were not the same as those presented to the ears of the listeners (the hearing aid was in between). I feel this is a missing link in the current version of the manuscript.

Lines 643-646 and conclusion: although the discussion of and reference to listening effort is relevant, this study does not allow interpretation of the results in terms of "listening effort" and listening difficulty. It is unknown to what extent and when the listeners actually tried to perceive the speech that was available. Therefore, it is impossible to conclude that the changes in HR are associated with listening effort.

Table 1: a direct comparison of the models including and excluding movement data is made. However, movement data were only available for 20% of the data points. A more fair comparison can be made by comparing the model based on the same $n = 5613$ observations with versus without movement data included.

Minor comments

Abstract line 20: investigate should be investigated

Line 84: remove causal

Line 99: the phrase "elements of active listening" seems odd

line 124: this should be thus

line 169: are should be were

lines 465-467: seems to repeat previous statements about the GVIF.

line 524: suggests should be suggest

Review form: Reviewer 3

Is the manuscript scientifically sound in its present form?

Yes

Are the interpretations and conclusions justified by the results?

Yes

Is the language acceptable?

Yes

Do you have any ethical concerns with this paper?

No

Have you any concerns about statistical analyses in this paper?

No

Recommendation?

Accept with minor revision (please list in comments)

Comments to the Author(s)

I feel this paper is much stronger as now written. Thank for diligently addressing my concerns. My two remaining minor suggestions was to: 1) work on the organization of the results, mainly by having the section headings more clear. I made suggestions on the appropriate pages in the results below. Also, it seems that HR needed its own section in results. Seemed like it was just lumped in with the data logging. 2) I questioned whether it was fair to say that the noise soundscape was more complex than speech in noise. I included my minor editorial comments below as well.

Abstract

Page 3

Line 27 "heart rate" instead of "heart rates" (my preference)

Line 29 perhaps better stated "In addition, and not previously recognized, increases in ambient sound quality- that is more favorable signal to noise ratios – are associated with decreases in mean heart rate.

Introduction

Revision is much improved.

Page 4

Line 41 delete "but" add period, new sentence Despite...

Line 44 delete "components, e.g. Flamme (1)" put (1) at end of sentence

Line 58 delete "or" after HR, add comma & substitute "and can" for "or" after "pressure (14),"

Page 5

Line 83 delete "Nevertheless" start sentence with "Besides"

Page 7

line 121 delete , after which

line 124 substitute "this" with "thus"

line 126 reword "Indeed, in a recent real-world study (27) increases in sound intensity over a seven day period were associated with...."

Line 132- Page 8 Line 135 ADD period after "parameters", add back in "Highlighting the relevance conducting real-world studies....ADD to that sentence "examining the effects of sound immersion on the human cardiovascular system (27,28)."

Line 137 Add "for example" after "city planning"

Line 144 HR instead of HRs

Page 10-11 Not sure on the headings used, suggest:

Acoustics

Variables collected

Seems like HR should have its own section but is lumped in with datalogging

Page 16

Not sure on headings again, suggest:

Descriptive statistics

Acoustics

Page 17

Line 340 reword "This supports the validity of the soundscape classification."

Page 19

Line 372

Descriptive statistics

Heart rate

Page 23

Line 451 substitute "sound wave" with acoustic signal

Line 459 delete "(i.e., modulated or not)"

Page 26

A question: Is noise more complex than speech in noise? How would you argue that?

Line 509-510 replace "different" with "the"

Discussion-Conclusion-Limitations

No suggestions or comments, well written

Decision letter (RSOS-201345.R1)

Dear Dr Christensen

On behalf of the Editors, we are pleased to inform you that your Manuscript RSOS-201345.R1 "The everyday acoustic environment and its association with human heart rate: evidence from real-world data logging with hearing aids and wearables" has been accepted for publication in Royal Society Open Science subject to minor revision in accordance with the referees' reports. Please find the referees' comments along with any feedback from the Editors below my signature.

Please submit your revised manuscript and required files (see below) no later than 7 days from today's (ie 12-Jan-2021) date. Note: the ScholarOne system will 'lock' if submission of the revision is attempted 7 or more days after the deadline. If you do not think you will be able to meet this deadline please contact the editorial office immediately.

on behalf of Dr César Lima (Associate Editor) and Pete Smith (Subject Editor)
openscience@royalsociety.org

Reviewer comments to Author:

Reviewer: 1

Comments to the Author(s)

Review of: "The everyday acoustic environment and its association with human heart rate: evidence from real-world data logging with hearing aids and wearables."

Although the readability and clarity of the revised version of this manuscript is improved, I still have several comments regarding the description of the results.

Main comments

page 4-5: Both subjective and objective measures of stress can provide valuable insight into the effects of noise and high task demand on stress. Moreover, especially the combination of such measures is useful (e.g. collecting subjective data in the present study would have been truly interesting as well). I suggest the authors change the conclusion of the first partial paragraph on page 5. Note that I refer to the version of the MS including track changes.

Lines 99-101: RSA is a measure primarily influenced by vagal tone. This is not the same as directly reflecting "sympathetic suppression". Please change the wording of this statement.

Results: still the results section is hard to follow. This is caused by not introducing the sections and analyses properly (in the statistical analysis paragraph: please describe which question will be answered by which analysis). For example, properly introduce section "moderating effect of soundscape". Also, a lot of statements and entire paragraphs include reflections on the (validity of) the results, comparison with previous studies and interpretation of the results. These should be moved to the discussion section. For example, lines 356-357, lines 368-372, line 363, first paragraph of page 19, line 406, line 411, lines 510-etc.

Figure 4 and interpretation of the results. All participants were hearing aid users (Oticon OpN hearing aids). This aspect should be discussed more thoroughly in the paper. It means that *despite* the use of this hearing aid, the results show the associations shown in Figure 4. As this advanced hearing aid will process and alter the sound characteristics, we may assume stronger associations in linear hearing aids. Furthermore, as the directional and noise reduction features of this hearing aid become more effective at higher sound levels, it seems remarkable that the relationship between SNR and HR is especially observed for higher SPL levels. Does this suggest that these effects of the hearing aid are marginal? Or were these turned off? Please put the results in the context of the hearing aid processing algorithm, as the sound characteristics as analysed in this manuscript were not the same as those presented to the ears of the listeners (the hearing aid was in between). I feel this is a missing link in the current version of the manuscript.

Lines 643-646 and conclusion: although the discussion of and reference to listening effort is relevant, this study does not allow interpretation of the results in terms of "listening effort" and listening difficulty. It is unknown to what extent and when the listeners actually tried to perceive the speech that was available. Therefore, it is impossible to conclude that the changes in HR are associated with listening effort.

Table 1: a direct comparison of the models including and excluding movement data is made. However, movement data were only available for 20% of the data points. A more fair comparison can be made by comparing the model based on the same $n = 5613$ observations with versus without movement data included.

Minor comments

Abstract line 20: investigate should be investigated
 Line 84: remove causal
 Line 99: the phrase "elements of active listening" seems odd
 line 124: this should be thus
 line 169: are should be were
 lines 465-467: seems to repeat previous statements about the GVIF.
 line 524: suggests should be suggest

Reviewer: 3

Comments to the Author(s)

I feel this paper is much stronger as now written. Thank for diligently addressing my concerns. My two remaining minor suggestions was to: 1) work on the organization of the results, mainly by having the section headings more clear. I made suggestions on the appropriate pages in the results below. Also, it seems that HR needed its own section in results. Seemed like it was just lumped in with the data logging. 2) I questioned whether it was fair to say that the noise soundscape was more complex than speech in noise. I included my minor editorial comments below as well.

Abstract

Page 3

Line 27 "heart rate" instead of "heart rates" (my preference)
 Line 29 perhaps better stated "In addition, and not previously recognized, increases in ambient sound quality- that is more favorable signal to noise ratios – are associated with decreases in mean heart rate.

Introduction

Revision is much improved.

Page 4

Line 41 delete "but" add period, new sentence Despite....
 Line 44 44 delete "components, e.g. Flamme (1)" put (1) at end of sentence
 Line 58 delete "or" after HR, add comma & substitute "and can" for "or" after "pressure (14),"

Page 5

Line 83 delete "Nevertheless" start sentence with "Besides"

Page 7

line 121 delete , after which
 line 124 substitute "this" with "thus"
 line 126 reword "Indeed, in a recent real-world study (27) increases in sound intensity over a seven day period were associated with...."
 Line 132- Page 8 Line 135 ADD period after "parameters", add back in "Highlighting the relevance conducting real-world studies....ADD to that sentence "examining the effects of sound immersion on the human cardiovascular system (27,28)."
 Line 137 Add "for example" after "city planning"

Line 144 HR instead of HRs

Page 10-11 Not sure on the headings used, suggest:

Acoustics

Variables collected

Seems like HR should have its own section but is lumped in with datalogging

Page 16

Not sure on headings again, suggest:

Descriptive statistics

Acoustics

Page 17

Line 340 reword "This supports the validity of the soundscape classification."

Page 19

Line 372

Descriptive statistics

Heart rate

Page 23

Line 451 substitute "sound wave" with acoustic signal

Line 459 delete "(i.e., modulated or not)"

Page 26

A question: Is noise more complex than speech in noise? How would you argue that?

Line 509-510 replace "different" with "the"

Discussion-Conclusion-Limitations

No suggestions or comments, well written

===PREPARING YOUR MANUSCRIPT===

===PREPARING YOUR REVISION IN SCHOLARONE===

Author's Response to Decision Letter for (RSOS-201345.R1)

See Appendix B.

Decision letter (RSOS-201345.R2)

Dear Dr Christensen,

It is a pleasure to accept your manuscript entitled "The everyday acoustic environment and its association with human heart rate: evidence from real-world data logging with hearing aids and wearables" in its current form for publication in Royal Society Open Science.

on behalf of Dr César Lima (Associate Editor) and Pete Smith (Subject Editor)
openscience@royalsociety.org

Appendix A

Author response

We thank the reviewers for the many helpful comments and suggestions, which we believe improves our manuscript significantly.

Below, we thoroughly address the reviewer comments point-by-point. The original review comments are in red and our response is in black. In addition, we have enumerated each reviewer comment so that cross-referencing can be made.

Besides addressing the reviewer comments, we have:

- Changed the aesthetics of the figures to improve readability.
- Re-fitted the statistical models since a minor mistake in the filtering of data meant that not all observations were included (no significant changes to the results).
- Edited the description of the data (total users and observations) to correct an erroneous description.
- Changed the title to specify the content of the paper more precisely.
- Changed “Sound exposure” to “Acoustic environment” for more precise terminology
- Removed ESM 1 since that information is now embedded into the manuscript (i.e. model fitting statistics).
- Fixed the author affiliation.

Reviewer: 1

Comments to the Author(s)

This manuscript describes the relationships between Heart Rate (HR) and several acoustical parameters and categories as extracted with a commercial hearing aid.

Although I very much appreciate the attempt to collect real-world data to address the research questions, I have several comments with respect to the methods and description of the methods.

Q1.1

First of all, more information about the methods is required. The authors could provide more details about the device (and data quality) used to collect heart rate data.

The devices for measuring heart rates are the user’s own wearables. Thus, it is a selection of smart watches (Apple watch, Garmin etc.) that all connect to the Apple Health app. We do not have access to the name of the devices used by the participants in the study. Comparing the data quality of devices from different manufactures is also beyond the scope of this study.

However, we do acknowledge that some form of data quality assurance is in order. We report descriptive statistics of the HR data and compare it to other studies that used medical-grade heart rate monitors, which we did following best practice for working with data from wearables (e.g. Hicks et al., 2019 – see *Methods*).

In addition, we added a paragraph to the discussion around this topic as well and have added references to investigations into the validity of data from consumer wearables in the Methods section (e.g. Hicks et al., 2019 and Witt et al., 2019). These studies highlight the potential for using consumer wearable data for observational studies. Please note that in the current study we are not concerned with absolute heart rates (e.g. for clinical diagnostic relevance) but only consider the association between changes in heart rate and sound.

Q1.2

Also, please provide more background in the introduction about the validity of HR as a measure of listening effort? Or stress? HR is an index of SNS activity but is also influenced by the PNS - please describe the underlying physiological mechanism. Changes in stress do not immediately reflect changes in HR and this relationship also differs between individuals.

We have added a paragraph to the introduction regarding listening effort and cardiovascular stress and have introduced the basic mechanism of sympathetic nervous system (SNS) and parasympathetic nervous system (PNS) impact on HR. Briefly, we describe the finding that task demands (i.e. listening condition) and motivation can modulate the listening effort (e.g. Pichora-Fuller et al., 2016) which in turn can modulate cardiovascular signals. Moreover, we have added how PNS typically affect stress levels and HR.

We do not wish to go into more specific physiological mechanisms as we in the current study only have access to mean heart rates – i.e. we cannot disentangle SNS and PNS contributions to HR. Please note it was never our aim to disentangle SNS and PNS effects of sound exposure. Instead we have expanded the discussion (“Limitations”) with statements regarding the potential disentangling of SNS and PNS using HRV.

Q1.3

Please discuss the limitations of the HR measure as collected in the daily-life conditions (i.e. no baseline, no posture correction).

In “Limitations”, we have added a paragraph stating the limiting factor of only considering mean heart rates in assessing cardiovascular stress. In addition, we discuss the limiting factor of using real-world data (i.e. lack of control).

Q1.4

The authors do correct for changes in GPS location at some point, but it is unclear how location information directly translates into movement (as participants could change location without moving (and without much effort) when in a bus or being a passenger in a car, and move /train without changing locations when exercising on their home gym station, for example).

When applying the LME models using a subset of data including movement activity (GPS), we found that the regression coefficients between sound and HR did not notably change (similar magnitude and highly overlapping 95% CIs), but that the addition of movement as a random factor increased the models ability to explain mean HRs. Thus, we can claim that effects of movement activity help in understanding HR variations but does not affect the association between sound and HR.

We acknowledge that the description of our applied movement correction is insufficient and have expanded it. Briefly, we compute an average speed (m/sec) in the time interval prior to each HR measure. If that speed is more than 10 m/sec (i.e. typical biking speed), we reject the data since most likely the activity is caused by passive transportation (i.e. car, train).

We have also updated the results section with a table (Table 2) of explained variance for all models. Thus, a direct comparison between models with and without movement correction can be made. As evident, adding movement as a random effect improves the explained variance of the full model but does not change the explained variance of the fixed effect predictors at least for the acoustic data model. For the soundscape model, it seems movement might have impacted the magnitude of the regression coefficient for contrasting “Speech in Noise” with “Noise”. Lastly, we have added a statement and the results of how the classified movement is linearly associated with increased HR from modeling it with a mixed-effect model, which was also expected.

Q1.5

The literature review mainly focuses on the effect of environmental noise on HR. However, the current

study aim seems more related to the effect of "high auditory task load" on HR, and not just the effect of any noise per se. Please make a clear distinction between these two aims in the literature review. Please also include more literature relevant to the actual current aim (ie studies on auditory task load using cardiovascular or other physiological biomarkers of effort and stress). As indicated by the current results and as suggested by literature from the same group (Lunner et al.), high noise levels are fairly uncommon in daily life. So the focus of the literature review needs to be adapted accordingly.

We thank and agree with the reviewer for pointing out that this distinction should be made. We have adjusted the introduction accordingly and have added more references to corroborate that listening conditions beyond sound intensity can affect cardiovascular systems. We added a paragraph to the introduction highlighting that active listening in noisy environments can also cause increased stress.

Q1.6

Furthermore, more detail is required a priori about the statistical analyses performed. Several analyses are performed, but they are not properly introduced in a statistical methods section. Therefore, the results are difficult to follow, and the connection between research questions and analyses is not always clear. Also, it would be helpful to clearly provide a description of the parameters analysed and the variables controlled for (like the location information).

We have expanded the statistical analysis section significantly to include a more in-depth description of the applied LME models and other tests performed. In addition, we have taken out the Pearson correlations assessing the mutual relationship between acoustic data.

Q1.7

Ethics: I'm not familiar with Danish law, but I find it remarkable that this study can be performed without informed consent procedure and some form of ethical approval. It would be appropriate to let an ethics committee test and confirm the statement of the authors ("no approval needed"). The authors could then refer to that decision in the ethics statement.

The study relies on anonymized data extracted from a database – such study does not require ethical approval according to the Danish Scientific National Ethics Committee (<https://www.nvk.dk/forsker/naar-du-anmelder/hvilke-projekter-skal-jeg-anmelde>). Informed consent was however given by the users – and we have added this previously neglected detail to the section.

Q1.8

More information is required regarding the selection of users. Were these regular hearing aid users selected from a pool who's data were available to Oticon? Or were they selected via hearing aid dispenser data? Please explain.

The data represents a convenience sample – that is, we extracted data from those users in the chosen time-window that had the most amount of data available. We have expanded upon the information in "Participants and Ethics".

Q1.9

The lack of information about the age of the listeners may be problematic in case the influence of auditory load interacts with age. It is known that age influences HR, so this is not unlikely. Please discuss.

In "Limitations" we discuss the potential effect of age on sensitivity towards listening environments.

Q1.10

Figures: Figures are difficult to read. The position of the labels on the axes is off, and the labels overlap. On greyscale printed paper, there is no visible difference between the colours used to indicate the categories etc. Figure citations are off as well (Figure 55 instead of 5 etc.).

We are sorry that the figures did not render probably in the PDF conversion of our manuscript. We have ensured that they are now clearly visible and rendered probably. Thanks for pointing out the erroneous figure reference – these are now fixed.

Q1.11

Page 18: Pearson correlations are used to assess the association between sound variables. I wonder whether this approach is the most appropriate - what about calculating ICCs?

We acknowledge that Pearson correlations are not appropriate. We merely wanted to “set the stage” for the LME modeling by conducting a prior assessment of the correlation between predictors. However, we have simplified and taken out the Pearson correlations as they are not important. Instead, we assess potential issues of multicollinearity by relying on the computed variance inflation factor (see Methods). This is a standard method for working with LME models and an alternative to the proposed ICC for diagnosing how correlated predictors in the LME model were. As evident, the highest variance inflation factor we find is 1.79, and the recommendation is that those should ideally be below 4.

Q1.12

Line 349: Speech in noise versus noise differ in a relevant dimension as the motivation of the listener may differ to a great extent between these conditions. Listeners may try to listen to the speech in noise (and may feel stressed when this is difficult or impossible). Alternatively, they may be immersed in a noisy condition without feeling stressed about this as they don't have to perceive any speech anyway (eg. when walking on a train platform). Hence, merging these two categories reduces the sensitivity of the analyses to detect meaningful effects.

We tend to agree with the reviewer in that more interesting interpretations can be made if the two soundscapes are not collapsed. This is a trade-off between ensuring fair statistical evaluations and being able to go in-depth with the research question. However, since we do not monitor the users listening intentions and activities, we cannot separate potential differences in stress levels due to e.g. “walking on a platform” versus “wanting to listen” in each soundscape. Thus, we have decided to keep the merging of the soundscapes since, as also evident in Figure 5, the acoustic data is highly overlapping between “Speech in Noise” and “Noise”. In a future project, it would be very interesting to examine the specific moderation of “Noise” and “Speech in Noise” on cardiovascular stress and sensitivity with a recording of specific listening intention and/or activities.

Q1.13

Lines 396-401: HR is also influenced by PNS activity. Please adjust this paragraph accordingly.

We have followed the suggestion and have deleted the statement that only the sympathetic branch is modulated by sound pressure. Instead, we state that ANS is moderated by SPL (in this study we do not know the SNS/PNS balance). Our previous assertion that the SNS branch of ANS is affected by SPL is based upon the studies referenced.

Q1.14

lines 57-59: please clarify whether the relationship cited was or was not statistically significant. The current phrasing is unclear.

We have rephrased the sentence as suggested

Q1.15

Line 184: a word is missing

Added

Q1.16

Hearing aid classification: This is a black box - no info is provided about the methods. Is there any supporting or background information about the method and validity of the classification algorithm in this user group in daily life conditions?

Each hearing aid manufacture apply their own proprietary sound classifiers. This is also true for Oticon hearing aids. Thus, we cannot disclose the full details of how the classifier operates. Instead, we can investigate from the data how well the classifier can cluster sound data (as in Figure 1 and 5) and use this as an indication of the classifier accuracy. In addition, previous studies have used such classified data to report on the daily sound environments faced by older hearing aid users (e.g. Humes et al., 2018), and in the current manuscript our descriptive findings using the classified soundscape corroborates well with previous findings. However, in the end, accuracy of a soundscape classifier is arbitrary – there is no objective threshold determining when a sound environment can be classified as “Speech”.

The main principles for the proprietary estimators have been published in a number of publications around 2004-2005 [1], and although this is not the full details we find that they have proven their applicability in controlling hearing aids for the benefit of their millions of individual users for a decade from back then. Also, many clinical audiologists have used individual distributions of these estimators accumulated in hearing aids and displayed in the fitting software in daily rehabilitation to get a more detailed picture of the individual hearing aid users. While we are unfortunately prevented from sharing additional details, we do find that the estimators with the current level of publicly available documentation is sufficient and quite relevant due to the widespread use.

[1] Chung, King. (2004). Challenges and Recent Developments in Hearing Aids Part I. Speech Understanding in Noise, Microphone Technologies and Noise Reduction Algorithms. Trends in amplification. 8. 83-124. 10.1177/108471380400800302.

Q1.17

Relevant literature: Studies performed by Holube et al. - they also asses relationships between acoustical featured and subjective effort and stress. von Gablenz, P., Kowalk, U., Bitzer, J., Meis, M., & Holube, I. (2019). Individual hearing aid benefit: Ecological momentary assessment of hearing abilities. In Proceedings of the International Symposium on Auditory and Audiological Research (Vol. 7, pp. 213-220).

We thank the reviewer for the suggestions. We have added the Holube et al., (2019) to the introduction as we found it very relevant.

Q1.18

Line 92: "were" should be "was"

Fixed

Q1.19

lines 112-113: please explain this further and provide relevant references for this statement.

We have added more description and two additional references

Q1.20

Line 158: what happens with the data of the additional 2 mins per 7 min period?

Each heart rate measure represents a 5-min running average. Thus, the heart rates within the 2-minute window is not included in the average.

Q1.21

lines 312-313: please use HR consistently (instead of pulse rate)

Thanks for pointing this out. Fixed

Q1.22

Line 313: SN should be SNR, I assume

Fixed

Q1.23

Line 316: were should be was.

Fixed

Q1.24

Having a native English speaker amongst the co-authors may be exploited a bit more to prevent these small typo's and grammatical errors :).

The native English co-author has performed a critical language review on the revised manuscript.

Reviewer: 2

Comments to the Author(s)

This manuscript left me with mixed feelings. I think that the main research rationale presented in the introduction section—sound characteristics beyond intensity may influence ANS activity and cardiovascular health—is interesting and original. I also liked the well-written introduction section that provides a nice overview of the existing literature and the straightforward description of the methods. Unfortunately, the results and discussion sections do not keep up with the high quality of the first two sections. Moreover, there also seems to be a mismatch between the rationale presented in the introduction and the described methods and results. In sum, I do not think that the manuscript warrants publication in its current form. A thorough revision might bring it above the publication threshold, but the main contribution of the manuscript would be descriptive information. Due to the lack of appropriate measures, it will not significantly advance the understanding of the effect of sound characteristics on ANS and cardiovascular health.

Please let me refer to my main concerns in more detail:

Q2.1

1) There is a mismatch between the employed methods and the research rationale presented in the introduction section. First, to demonstrate that sound characteristics influence cardiovascular health, one would obviously have to assess cardiovascular health. The presented work failed to do so.

We would like to stress that it was never our intent to assess cardiovascular health, and we also did not claim that in the manuscript. In the introduction we specifically write that we assess longitudinal data to assess how everyday heart rates are modulated by sound exposure. The introduction merely serves to highlight the importance for understanding these everyday associations between sound and cardiovascular reactions since they have implications for e.g. cardiovascular health.

To make our message more concise, we have adjusted the introduction so that less emphasis is put on cardiovascular health aspects.

Q2.2

Second, the introduction nicely points out that it is mainly sympathetic activity that has been associated with poor health outcomes and it also shows that preceding work on this topic has differentiated between sympathetic and parasympathetic responses. Given the differentiated view offered in the introduction, I was surprised to read that the presented work only assessed heart rate as outcome. Heart rate does not allow the separation of sympathetic and parasympathetic effects and does thus constitute a step backwards in relation to preceding work on the topic.

I understand that there might have been practical constraints that prevented the authors from assessing health outcomes and specific indicators of sympathetic and parasympathetic ANS activity, but this does not resolve the fundamental issue: To add to and extend the research discussed in the introduction section, one needs to assess specific indicators of sympathetic (and parasympathetic) activity and collect data related to cardiovascular health.

As the reviewer points out, we are limited to only inspecting mean heart rates, which means we cannot disentangle SNS and PNS activity directly. However, our point to make in the introduction was to say that the traditional investigations of the effect of sound *intensity* on heart rates might not be sufficient since literature shows that sound *type* (i.e. complexity, noise content) can also affect HR and presumably so by the PNS branch. Thus, in ecological studies, we also need to consider such sound dimensions. We can then speculate if our documented negative association between HR and SNR is in fact due to elevated PNS activity or from withdrawals of SNS activity – this is certainly something we will follow up on in future research.

Q2.3

2) It is difficult to find in the results section the crucial tests that addressed the main research aim. I can understand the wish to provide a comprehensive description of the collected data, but the results section presents so much descriptive information that it is difficult to locate the most important information—the statistical tests that demonstrate the impact of SPL, SML, SNR, and sound scape on HR. I would recommend that the authors revise the results section to provide more guidance and make more salient what the important information is.

We have followed the reviewer's advice and adjusted the results section accordingly with text guiding towards the main result. However, we do believe a thorough description of the data is necessary due to its real-world nature and novelty it.

Q2.4

3) The authors' conclusions sometimes seem to be based on visual inspection of graphs or the interpretation of the absolute numbers instead on inferential tests (see for instance the last sentence of the first paragraph on page 17 or the sentence beginning with "The most correlated parameters..." In the first paragraph on page 18). The authors should revise their results and discussion sections to make sure that all conclusions are backed up by appropriate statistical tests.

We have followed the reviewer's advice and adjusted the results section accordingly. For example, Figure 4 now include statistical tests for the presented slopes in the figure caption and the Pearson correlations the reviewer refers to have been removed.

Q2.5

4) Given the largely exploratory testing strategy, I am concerned about alpha (type-I) error inflation. The authors used alpha error control strategies for some of their post-hoc comparisons, but they did not control for the overall high number of tests that they conducted. I would recommend to either limit p-value based testing to tests that can be justified by a hypothesis or to employ an alpha error control strategy that takes into account the total number of p-value based tests that were conducted.

We disagree with the reviewer that Type-I errors should be a problem for the current results. We primarily employ mixed models in our analysis, because mixed models are vastly superior in controlling for Type I errors than alternative approaches and consequently results from mixed models are more likely to generalize to new observations (e.g., Barr, Levy, Scheepers, & Tily, 2013 *Journal of Memory and Language*). Also, the testing strategy was not exploratory. We employed the mixed-models with clear prior hypotheses based on the previous findings listed in the introduction. The remaining tests (ANOVA with Bonferroni correction and Likelihood-ratio tests between different mixed-models) were likewise strongly based on hypotheses such as "adding a predictor to a model increase the models fit to data".

We have removed the Pearson correlation tests and p-values for assessing the association between acoustic data predictors (see our response Q1.9 to reviewer 1).

Q2.6

5) P-value based testing following a Neyman-Pearson approach requires a careful determination of sample size using an a priori power analysis. I would appreciate if the authors could include information on how they determined the sample size of the presented study.

We have removed the Pearson correlation tests and p-values for assessing the association between acoustic data predictors (see our response Q1.9 to reviewer 1). Instead, we assess potential multicollinearity by computing variance inflation factors (see Methods).

Q2.7

6) I was surprised to read that the authors expected heart rate to show a uniform distribution (second sentence on page 10). It was the first time that I read that all heart rate values should be equally likely.

Most papers that I read assumed a normal distribution for heart rate. I also think that some of the statistical procedures that the authors employed to not go well with uniformly distributed data. Moreover, I was wondering whether the authors also had this expectation of uniformly distributed data for the sound parameters that they analysed and whether they applied the same outlier exclusion procedure to these variables.

We do not assume heart rates to be uniformly distributed. Rather, we assume they should be somewhat normally distributed. This is also why we chose to filter away data below the 5% and above the 95% quantile so that these extreme and low-incidence heart rates do not impact the statistical modeling and we could apply the simpler linear mixed-effect model. We back this up by comparing the distribution of residuals before and after filtration (appendix), which was the reason for why we in the first place needed to do active filtering. The filtration only affected the numerical value of the regression coefficients but not the significance and order – we also note this in the manuscript.

We have changed the wording to avoid this confusion.

We did not do the same with the independent variables since they do not affect the residuals.

Q2.8

7) In my pdf version of the manuscript, the readability of some of the figures was low due to overlapping text.

Apologies for this and we have ensured a better quality of the figures in the revised manuscript

Reviewer: 3

Q3.1

Comments to the Author(s)

My general comments are that I enjoyed the paper but thought:

1) the presentation of the statistical analyses and the reasoning behind them needed to be clearer. This includes variance explained in the regression analyses. Also I saw the ANOVA results for soundscape but did not see the statistics for the raw SPL, SML and SNR. It seems like you meant to include that?

We have cleaned up the statistical analysis section to better convey the underlying idea – i.e. that associations between sound characteristics and HR are being investigated with mixed-effects models. In addition, we have expanded upon the description of other analyses such as the variance inflation factor and explained variance.

In fact, we appreciate the reviewer's suggestion for adding more effect size type of information. We have computed explained variance following recommendations for multilevel models (see Statistical Analysis under Methods). Thus, we report on the explained variance from the fixed effects alone as well as from the full models including random effects.

Unfortunately, due to the way that variance is partitioned in linear mixed models (e.g., Rights, J. D., & Sterba, S. K. (2019). *Psychological Methods*), there does not exist an agreed upon way to calculate standard effect sizes for individual model terms such as main effects of each acoustic data characteristic or interactions. For individual model terms, we report unstandardized effect sizes which is in line with general recommendation of how to report effect sizes (e.g., Pek & Flora, 2018, *Psychological Methods*), and have added a more thorough discussion of coefficient magnitudes in relation to expected change in HR with sound. Specifically, we have re-fitted the models after log-transforming the outcome (heart rates), which means we can directly compare our regression coefficient effect sizes with those of previous and similar work. As evident, the coefficients we find are highly comparable to those from others, pointing towards a more robust real-world effect size in terms of expected change in HR with change in sound level. We have added these details in a subsection in the results called "Effect size considerations"

The ANOVA for soundscapes were introduced as a first step assessment of the data presented in Figure 3, which is then followed up by a more rigorous statistical modeling in Table 1. Since SPL, SML, and SNR are continuous we do not apply an ANOVA. Instead, we have added statistical evaluation of the regression slopes in Figure 4 as a first step assessment of the association between raw acoustics and HR. Again, this is then followed up by detailed statistical modeling in Table 1

2) More discussion on HR and HRV in the introduction and discussion section is needed.

We have added more regarding HRV in the discussion, and have expanded the introduction with more on HR. Note that since we do not investigate HRV in the current study, we would like to refrain from spending too much time on that measure in the introduction.

3) More on why SML is relevant in introduction.

We have expanded a bit on the definition of SML in the introduction and describe in the discussion how the association between SML and HR are to be interpreted. Briefly, the introduction brings about the fact that more dimensions than just sound intensity is needed to evaluate the cardiovascular impact – hence, SML and SNR is introduced.

Q3.2

Introduction

Page 6

Line 89-90 Not sure on sentence saying concomitant increase in HR and HRV values as many HRV values typically decrease with increases in HR

Please add more discussion regarding HRV

We merely reproduce the reported findings from the cited study – i.e. they found increases in both HR and HRV immediately following increases in sound intensity, but that HRV then subsequently decreased as a delayed response. We have added an additional note about HRV and SNS/PNS as well as more on HRV in the discussion (under “Limitations”) as the reviewer suggests.

Q3.3

Methods

Page 7

Do the wearables provide any additional data than HR?

The data in the current study only included heart rates from the wearables.

Q3.4

Page 8

Line 129 Define wearables. Are these all watches?

The wearables are all devices that continuously measure heart rates and are linked to the Apple Health app on iOS. To the authors knowledge, such devices are considered “smart-watches” – e.g. apple watch, Garmin wristbands etc. We have added a short statement of this in the methods section together with references regarding the promising use of health data from wearables in observational studies. Also see our response to reviewer 1 (Q1.1)

Q3.5

Line 140 Define SML acronym. Also, why do we care about SML? Obvious why we care about SNR and SPL, not as clear about SML. I assume SML will typically be highest in conditions where there is speech or music and may be relevant to the HA digital noise reduction.

It is not obvious why we should care about SML – however, SML represents a dynamic dimension of the sound environment that has not been investigated yet and, given the information in the introduction, might correlate with cardiovascular signals. In the discussion we try to give our take on why SML might correlate positively with HRs – e.g. changes in SML might be related to active listening tasks with speech or music that elevate the listening effort exerted.

Q3.6

Page 10

Line 175 Do you have another source that used similar exclusion criteria? Why exclude the lowest 5 and upper 5%. Was it effective in removing not only low incident HR but also artifact? Also, what was done if anything to remove artifact?

It is customary to ensure that the statistical models adhere to assumptions of e.g. residual homogeneity. In our case, this meant that we had to exclude extreme values. As you can also see in the appendix, removing these extreme values did help the model in producing residuals that were normally distributed – hence, validating the statistical model. We do note that exclusion of the extreme values did not change the significance of any of the modeling outcomes.

We did not apply additional artefact removal. But we also do not expect that any artefacts are present in the data.

Q3.7

Page 11

How was movement data specifically used in analyses of HR?

Movement was added as a random-effect term to a subset of the data that included it. This means, that if the changes in HR were caused by movement, we should see close to zero magnitude regression coefficients of the sound data predictors. This was clearly not the case – in fact, adding movement did not affect the coefficients for the acoustic data model at all while only affecting the soundscape model in the contrast between “Speech in Noise” and “Noise”. This indicates that the regression of sound and HR is not caused by a latent effect of physical movement. We expanded upon the Methods section to better explain the logic behind this.

Q3.8

Line 206 What does “nlme” stand for? I assume non-linear and linear mixed effects models

Yes. In this context it is the name of the R package we used. We have added some more details to ensure replicability of methods can be achieved.

Q3.9

Results

Page 12

Define evening, late at night, early in morning.

Fixed.

Q3.10

Page 13-14

Strange wording related to stating that the results here are consistent with other studies using similarly aged participants. HR does not indicate or determine age (perhaps translation issue here?)

We have reworded this sentence. We argue that the mean HR and SD in our data is much closer to real-world norms when only considering individuals older than 71 and younger than 80 years of age compared

to also including younger age groups.

Q3.11

Page 14

Line 248 I thought it was 5:00-24:00 before?

Fixed. We have ensured that data and text reflect that we only consider data between 6:00 and 24:00.

Q3.12

Page 15

Line 261 Figure 3 not 33

Fixed, thanks.

Q3.13

Page 16

Having trouble finding statistics indicating the relationship between HR and SNR, SML and SPL aside from those just looking at differences for HR between soundscapes. I thought that was described in methods.

We test the relationship between HR and SNR, SML and SPL with the linear mixed-effect model and the outcomes are presented in Table 1. In case the reviewer is referring to the marginal means (Figure 4) then this figure is meant to visualize a pooled relationship, which is then backed up by the statistical modeling (Table 1) of within-participant effects.

Q3.14

Figure 4, I am guessing indicates the relationships but does not give any statistics regarding the relationships

As above, the statistics for the relationships are given in Table 1. Figure 4 merely displays the relationships on aggregated levels so that the reader can follow the results better. Given the confusion, we have also added the results of testing if the fitted slopes in Figure 4 are significant in the Figure 4 caption. Note that, the slopes in Figure 4 represents the relationship between HR and SPL, SML, and SNR on an aggregated level – that is, without adjusting for individual sensitivity, time-of-day, and weekday. They are meant to complement the results listed in Table 1.

Q3.15

Line 276-280

Need more explanation of what bins and marginal means are and why they were used. What exactly is a bin center quantile in terms of how that relates to raw SPL, SNR and SML?

We followed the reviewer's advice and expanded the description of the marginal means and the computation thereof.

Q3.16

Line 293 Once again need to tell why we care about SML

We introduce SML in the introduction and go into details in the discussion as to why SML might modulate HR. We would like to refrain from adding such details in the results section since we believe such discussion belongs in the Discussion.

Q3.17

Page 18

Line 301 Figure 5, not Figure 55

Not sure the discussion on which factors were more correlated is appropriate or the phrase “most correlated parameters”, typically just talk about which were significant.

We test the correlation between predictors prior to entering them into the LME models. This is to describe to what extent collinearity between predictors are to be expected. In this sense, the magnitude of correlation is important - not whether it is significant or not – in our case with so many data-points even a small correlation is significant. However, in the revised manuscript, the correlation analysis is taken out since it does not contribute with additional information.

Q3.18

Page 19

Line 314. Not sure why any of this is important, explain offsets and intercepts.

We have removed this information since it is already explained in the Methods section under “Statistical analysis”. In addition, we have simplified the description of the stated random offsets (i.e. intercepts) and slopes in the Methods section.

Q3.19

Page 20

Statistics table seems like it is missing other relevant information above and beyond regression coefficients and CIs. Particularly for the acoustics data model. Such as explained variance, etc. You gave the information that movement, when included explained 32.8% more HR variance, please give values for this for all rows in table with and without movement. I think that seems some of the most important information.

See above response to Q3.1. In addition, we refrain from using estimated p-values for the coefficients but instead presents the 95% CIs.

Line 326 Reword sentence, start with: Although adjusted vs. non-adjusted for movement models had similar and overlapping confidence intervals, adjusted models explained an additional 32.8% of HR variance.

We have followed this advice

Q3.20

Page 21

Line 335-337. Be more precise. You are talking about the regression coefficient size and signage I believe.

We have specified this in more precise terms.

Q3.21

Page 22

Line 358 What exactly is a variance inflation factor and how does one interpret 1.79 as being small?

The variance inflation factor tells you how much variance of a regression coefficient is inflated due to multicollinearity (correlation between predictors) in the model. Thus, low variance inflation factors indicate that predictors are more independent and, thus, their estimated value and error are more certain. We have added a reference to this, expanded the description in the methods section, and added a short statement that a value 1.79 is typically deemed accepted in statistical modeling.

Q3.22

Line 368-369 when the soundscapes were more optimal for listening.....not sure how that is defined

We agree with the reviewer that the formulation was not precise. When the ambient sound environment contains little background noise, any speech-like signals can be clearly perceived. Also, if only

one speaker is present in contrast to a group, that one speaker's speech can be better understood and perceived. Thus, regarding the hearing aid classification of soundscapes, "Quiet" and "Speech" are optimal for listening to speech-like sources. We have changed the wording: "...when the ambient acoustic environment was classified as being favourable for listening (i.e. in quiet and speech-dominated soundscapes)"

Q3.23

Page 24

More discussion on heart rate literature

We have added a section in the discussion where we compare our findings of %-change in mean HR per change in sound level with that reported in the literature. Besides this, we are not sure what discussion the reviewer is referring to.

Q3.24

Line 402. Does SML simply indicate when speech or music was present? Quiet and noise conditions may at times have neither speech or noise signals, hence lower SML. Listening to speech and music would be cognitively engaging and may increase HR.

We cannot go to that level of detail with the current data. We can say that most likely, SML is an indicator of the presence of speech and music like signals – but it could also be a car alarm for example.

Appendix B

Dear Editor,

We once again thank the reviewers for their insightful comments. We believe addressing the concerns raised has improved the readability of our manuscript. Below, we address the reviewer comments point-by-point (in red).

In addition, we have made the following minor changes to the manuscript:

- Figures: All figures have updated colour schemes to improve readability for colour-blind (i.e. following suggestions from the author guidelines).
- Figure 2: Panel a-c is updated by expanding the data into quartiles for better transparency.
- Added Acknowledgement section before the references.
- Added a relevant reference to a paper published in December 2020 in the discussion.
- Changed “user” to “participant”

Reviewer comments to Author:

Reviewer: 1

Comments to the Author(s)

Review of: "The everyday acoustic environment and its association with human heart rate: evidence from real-world data logging with hearing aids and wearables."

Although the readability and clarity of the revised version of this manuscript is improved, I still have several comments regarding the description of the results.

Main comments

page 4-5: Both subjective and objective measures of stress can provide valuable insight into the effects of noise and high task demand on stress. Moreover, especially the combination of such measures is useful (e.g. collecting subjective data in the present study would have been truly interesting as well). I suggest the authors change the conclusion of the first partial paragraph on page 5. Note that I refer to the version of the MS including track changes.

We have changed the partial conclusion to state that objective measures should support the subjective reports. It now reads:

“Thus, to fully understand the impact of the everyday acoustic environment on the human body, subjective reporting of stress needs to be supported by objective measures such as cardiovascular reactions toward changes in the environment.”

Lines 99-101: RSA is a measure primarily influenced by vagal tone. This is not the same as directly reflecting "sympathetic suppression". Please change the wording of this statement.

In the referenced study, the authors (Umemura M & Honda K, 1998) use RSA to assess para-sympathetic activity and MWSA (Mayer wave related Sinus Arrhythmia) to assess both para- and sympathetic activity levels. Thus, they measure whether listening to music induces a stress reaction, and if so, whether the reaction is driven by increase/decrease in either sympathetic or para-sympathetic activity. They find that listening to classical music does not change either the sympathetic or para-sympathetic activity levels. On the other hand, listening to rock and noise increase sympathetic activity levels and decrease para-

sympathetic activity levels. The authors then argue that classical music effectively “suppress” sympathetic activity since it does not increase, and para-sympathetic activity levels does not change. We have re-worded the paragraph (see below) to clarify this issue and re-worded “sympathetic suppression” to “suppression of a sympathetic stress reaction”:

“Umemura M. & Honda K. (20) found that listening to classical music suppressed a sympathetic stress reaction (i.e. by keeping a stable ANS balance) compared to rock music or noise – both of which increased activity in the sympathetic control (measured by Mayer Wave related Sinus Arrhythmia) while simultaneously decreasing activity in the parasympathetic control (measured by Respiratory Sinus Arrhythmia). Moreover, the degree of suppression was positively correlated with subjective reports of comfort (20). “

Results: still the results section is hard to follow. This is caused by not introducing the sections and analyses properly (in the statistical analysis paragraph: please describe which question will be answered by which analysis). For example, properly introduce section "moderating effect of soundscape".

We now formally introduce the LME models with separate subheadings in the Statistical analysis section. Moreover, we have added subheadings for “Model diagnostics” and “Effect size estimation” to the section to better separate the different parts of the statistical modelling section.

In addition, the headings in the “Results” are changed (also following advice from reviewer 3) to improve coherence and flow while subsections in “Results” are started with a short introduction for clarity.

Also, a lot of statements and entire paragraphs include reflections on the (validity of) the results, comparison with previous studies and interpretation of the results. These should be moved to the discussion section. For example, lines 356-357, lines 368-372, line 363, first paragraph of page 19, line 406, line 411, lines 510-etc.

We agree that some statements should be moved to the discussion. We have moved the discussion about the impact of movement on the models and the interpretation of it (i.e. starting line 510 in the previous manuscript version the reviewer refer to).

However, some of the statements, especially in the section “Descriptive statistics”, are used to validate our data – i.e. the direct comparison to other findings in the literature is what comprise the result and we therefore prefer to keep those statements in the “Results” section. We then follow it up with a short statement in the “Discussions” that the validity of the data is proven since it highly resembles that of previous research. We do not wish to include this information in the discussion section because a comparison between our sound data and previously reported sound data is not the main aim of this study. To explain this, we have added a short statement at the beginning of the “Descriptive statistics” section.

Figure 4 and interpretation of the results. All participants were hearing aid users (Oticon OpN hearing aids). This aspect should be discussed more thoroughly in the paper. It means that *despite* the use of this hearing aid, the results show the associations shown in Figure 4. As this advanced hearing aid will process and alter the sound characteristics, we may assume stronger associations in linear hearing aids. Furthermore, as the directional and noise reduction features of this hearing aid become more effective at higher sounds levels, it seems remarkable that the relationship between SNR and HR is especially observed for higher SPL levels. Does this suggest that these effects of the hearing aid are marginal? Or were these turned off? Please put the results in the context of the hearing aid processing algorithm, as the sound characteristics as analysed in this manuscript were not the same as those presented to the ears of the listeners (the hearing aid was in between). I feel this is a missing link in the current version of the manuscript.

We agree that the fact that the participants in the current study wore hearing aids should be discussed. We have added the following paragraph in the “Limitations” to highlight this fact:

“Further, hearing aids process ambient sound with the goal of improving the SNR by changing how signals are amplified and noise is reduced. Detailed information about these parameters under varying real-world conditions are unavailable in our data. Thus, the effective SNR might differ from the logged ambient SNR presented in Figures 1 to 5. It is thus possible that the findings presented in Figure 4(e) and Table 1 would differ among people with normal hearing and/or with unaided impaired hearing. However, as noted in the introduction, laboratory testing of people with unaided hearing impairment also shows negative associations between the SNR of listening tasks and stress reactions measured as electrodermal activity (23)“

Lines 643-646 and conclusion: although the discussion of and reference to listening effort is relevant, this study does not allow interpretation of the results in terms of "listening effort" and listening difficulty. It is unknown to what extent and when the listeners actually tried to perceive the speech that was available. Therefore, it is impossible to conclude that the changes in HR are associated with listening effort.

We do not conclude that lower HR in high SNR conditions are associated directly to effects of listening effort, instead, we state that our finding suggests that real-world conditions like those used in previous laboratory work supports a decrease in listening effort (i.e. from higher SNR), which results in lower HR in fairly loud environments (i.e. >60 dB SPL) and soundscapes classified as ideal for listening (i.e. “Quiet” and “Speech”).

We have added a sentence in the discussion to highlight that we can only speculate about whether the effect is driven by listening effort. Future studies including subjective reporting of listening intentions will help answer such questions:

“However, future studies should include subjective reporting of listening intentions and experiences to reveal specifically the contribution of listening effort and fatigue to changes in everyday ANS activity. For example, experience sampling methods such as Ecological Momentary Assessments for assessing everyday-life listening experiences in hearing aid users (59-61) could be expanded with physiological monitoring.”

Table 1: a direct comparison of the models including and excluding movement data is made. However, movement data were only available for 20% of the data points. A more fair comparison can be made by comparing the model based on the same n = 5613 observations with versus without movement data included.

We agree and have added two columns to table 1 with the regression coefficients for the Soundscape and acoustic data model for the subset of the data with movement information but without adjusting for movement. Briefly, the coefficient magnitudes of the adjusted and non-adjusted models do not differ.

Minor comments

Abstract line 20: inverstigate should be investigates

Corrected

Line 84: remove causal

We disagree with this. The results summarized here are based on a laboratory manipulation of the sound intensity of noise exposure. This means the effect, i.e. a change in mean HR with a change in noise intensity, must be causal. We have therefore chosen to keep the word “causal” in the manuscript to highlight the fact that from laboratory tests at least, there is evidence that sound exposure has a direct causal effect on physiology.

Line 99: the phrase "elements of active listening" seems odd

We have changed it to "active listening".

line 124: this should be thus

Corrected

line 169: are should be were

Corrected

lines 465-467: seems to repeat previous statements about the GVIF.

We prefer to keep this short re-cap of how what the GVIF number can help with for better readability

line 524: suggests should be suggest

corrected

Reviewer: 3

Comments to the Author(s)

I feel this paper is much stronger as now written. Thank for diligently addressing my concerns. My two remaining minor suggestions was to: 1) work on the organization of the results, mainly by having the section headings more clear. I made suggestions on the appropriate pages in the results below. Also, it seems that HR needed its own section in results. Seemed like it was just lumped in with the data logging. 2) I questioned whether it was fair to say that the noise soundscape was more complex than speech in noise. I included my minor editorial comments below as well.

We have now addressed this input – see below for details.

Abstract, Page 3

Line 27 "heart rate" instead of "heart rates" (my preference)

Corrected

Line 29 perhaps better stated "In addition, and not previously recognized, increases in ambient sound quality- that is more favorable signal to noise ratios – are associated with decreases in mean heart rate.

Corrected

Introduction

Revision is much improved.

Page 4

Line 41 delete "but" add period, new sentence Despite....

corrected

Line 44 44 delete "components, e.g. Flamme (1)" put (1) at end of sentence

corrected

Line 58 delete "or" after HR, add comma & substitute "and can" for "or" after "pressure (14),"

corrected

Page 5

Line 83 delete “Nevertheless” start sentence with “Besides”

corrected

Page 7

line 121 delete , after which

corrected

line 124 substitute "this" with “thus”

Corrected

line 126 reword “Indeed, in a recent real-world study (27) increases in sound intensity over a seven day period were associated with....

Corrected

Line 132- Page 8 Line 135 ADD period after “parameters”, add back in “Highlighting the relevance conducting real-world studies....ADD to that sentence "examining the effects of sound immersion on the human cardiovascular system (27,28).”

Corrected

Line 137 Add “for example” after “city planning”

Corrected

Line 144 HR instead of HRs

Corrected

Page 10-11 Not sure on the headings used, suggest:

Acoustics

Variables collected

Seems like HR should have its own section but is lumped in with datalogging

We have re-organized the sections so that the main heading is “Variables collected” and then have added subheadings “Sound data” and “Heart rate data”

Page 16

Not sure on headings again, suggest:

Descriptive statistics

Acoustics

We agree and have re-organized the headings so that the main heading is “Descriptive statistics” and then have added subheadings “Soundscape and acoustic data” and “Heart rate data”

Page 17

Line 340 reword “This supports the validity of the soundscape classification.”

Corrected

Page 19

Line 372

Descriptive statistics

Heart rate

Corrected (see answer above)

Page 23

Line 451 substitute "sound wave" with acoustic signal

Corrected

Line 459 delete "(i.e., modulated or not)"

Corrected

Page 26

A question: Is noise more complex than speech in noise? How would you argue that?

We assume that both "Speech in Noise" and "Noise" represent complex (i.e. hard to listen in) listening conditions. It is true that it is a stretch to claim that "Speech in Noise" is more or less complex than "Noise" since we do not know what the actual listening intention was. However, the common objective metric for determining "complexity" of a soundscape is that the SNR and SML levels are low while the SPL is high. Thus, from visual inspection of Figure 5 "Noise" is most complex closely followed by "Speech in Noise". It can also be argued that something classified as pure Noise will always be more complex than something classified as containing a modulating signal (i.e. Speech in Noise).

We have added a brief definition about "complexity" of a soundscape the first time it appears in the manuscript (in "Results"):

"Note that complexity is defined as the interaction between SNR and SPL. High complexity is assigned to soundscapes with low SNR and high SPL (i.e. "Noise", see Figure 5)"

Line 509-510 replace "different" with "the"

Corrected

Discussion-Conclusion-Limitations

No suggestions or comments, well written